# Exploring the Learning Mechanisms of Neural Division Modules

**Bhumika Mistry**                                                           *bm4g15@soton.ac.uk*
*Department of Vision Learning, and Control*
*Electronics and Computer Science*
*University of Southampton*

**Katayoun Farrahi**                                                         *k.farrahi@soton.ac.uk*
*Department of Vision Learning, and Control*
*Electronics and Computer Science*
*University of Southampton*

**Jonathon Hare**                                                            *jsh2@ecs.soton.ac.uk*
*Department of Vision Learning, and Control*
*Electronics and Computer Science*
*University of Southampton*

**Reviewed on OpenReview:** *https://openreview.net/forum?id=HjelcW6wio*

## Abstract

Of the four fundamental arithmetic operations (+, -, ×, ÷), division is considered the most difficult for both humans and computers. In this paper, we show that robustly learning division in a systematic manner remains a challenge even at the simplest level of dividing two numbers. We propose two novel approaches for division which we call the Neural Reciprocal Unit (NRU) and the Neural Multiplicative Reciprocal Unit (NMRU), and present improvements for an existing division module, the Real Neural Power Unit (Real NPU). In total we measure robustness over 475 different training sets for setups with and without input redundancy. We discover robustness is greatly affected by the input sign for the Real NPU and NRU, input magnitude for the NMRU and input distribution for every module. Despite this issue, we show that the modules can learn as part of larger end-to-end networks.

## 1 Introduction

Division is one of the four fundamental arithmetic operations and is necessary for expressing real-world dynamical systems (Sahoo et al., 2018a) or physics-based formulas (Udrescu & Tegmark, 2020). However, the properties of division of values around zero leads to undesirable gradients for training Neural Networks (NN) through backpropagation, making it the hardest operation to learn (Mistry et al., 2022). Networks which try to learn division naively such as multilayer perceptrons (MLPs) are unable to deal with the fluctuant gradients caused by the asymptotic nature and discontinuities in division (Trask et al., 2018). Furthermore, if the network lacks an appropriate bias for learning division, it can also lead to poor generalisation on out-of-distribution (OOD) data. For example, a network could learn to achieve a reasonable loss on a training/validation range between $[1, 2]$ but be unable to maintain a reasonable loss when tested outside of $[1, 2]$ such as range $[3, 10]$.

In particular, imagine you must learn to divide 2 numbers from a list of 10 numbers, but are only given the 10 numbers and the expected result. This task requires finding the 2 relevant operands, the order to divide the operands, and learning to divide. In machine learning, this is equivalent to a supervised regression task

where the aim is to learn the underlying function between the inputs and output such that the solution is generalisable to any input. For NNs, the main challenge of this task comes from learning the selection and operation at the same time, which can lead to conflicting priorities when learning weights.

Selecting relevant inputs/features is a desirable property of neural networks useful for improved interpretability, reduced pre-processing costs and greater generalisation (Chandrashekar & Sahin, 2014). In particular, there exists a class of NNs called Neural Arithmetic Logic Modules (NALMs) which learn to select features and learn arithmetic operations simultaneously (Trask et al., 2018; Mistry et al., 2022). NALMs are designed to provide systematic generalisation for arithmetic operations such that if the appropriate parameters are learnt once training ends the NALM can extrapolate to unseen OOD data. Furthermore, as differentiable specialist modules (such as those for arithmetic operations) can be integrated with overparametrized networks as an intermediate module, being able to successfully select only the relevant inputs is important (Madsen & Johansen, 2020). However, even recent models still struggle to learn division when there is input redundancy (Schlör et al., 2020). Can we build models which can learn division in the presence of its undesirable, yet valid, properties?[1] We aim to address this question in this paper. Specifically, we contribute the following:[2]

- We show how additional biases improves learning of an existing division module, the Real NPU (Heim et al., 2020), by including: clipping, discretisation and constrained initialisation.

- We develop two novel division modules, the NRU and the NMRU, by extending an existing multiplication unit. Through rigorous experiments we find both modules outperform the Real NPU for the no input redundancy tasks, with the NMRU also outperforming the Real NPU for the input redundancy tasks.

- We identify the types of data which hinders learning division for each module, including training on: mixed-sign inputs, negative ranges, extremely small values and different distributions. These difficulties can be sufficiently identified using synthetic division tasks.

- We show how NALMs can learn in a larger end-to-end network using an arithmetic MNIST task.

## 2 Related Work

Learning to robustly divide provides a stepping stone for NNs in achieving symbolic regression. Symbolic regression searches the space of expressions to predict a mathematical expression from a given set of input-output observations. Compared to black-box NN functions, expressing a mathematical function is significantly easier to interpret. Symbolic regression can be implemented via Genetic Programming (GP) using Evolutionary Algorithms (EA) which learn mathematical expressions (Koza, 1994; Schmidt & Lipson, 2009). EAs maintain a population of expressions where individuals of the population get selected based-off a fitness function and modified via techniques such as crossover and mutation. This procedure is repeated until a stopping criterion is met. Hybrid methods which combine GP with local search can also be used to further boost generalisation results (Kommenda et al., 2020), however both the pure EA and hybrid methods do not scale well due to the combinatorial nature of the method.

Alternatively, a fully differentiable NN approach can be taken by incorporating biases to improve the interpretability of the network. Sahoo et al. (2018b) sets activation functions in a layer to different symbolic operations rather than using a traditional non-linear activation like ReLU. They also encourage only using relevant weights through a sparsity regularisation scheme which varies in strength depending on how much training has occurred. However, to gain the best performance requires using selection strategies over many trained modules which is costly and can be unreliable (Sahoo et al., 2018a). In contrast, Udrescu & Tegmark (2020) exploits patterns in the data by designing physics related biases such as transnational symmetry or multiplicative separability into their architecture. Due to the strong prior which assumes the dataset contains an underlying physics representations, the model performs poorly when trained on datasets without such representations (Cava et al., 2021).

---

[1] A desiderata for building a division module is provided in Appendix A.

[2] Code (MIT license) available at: https://github.com/bmistry4/nalm-division.

Another type of differentiable NNs are NALMs which have specially designed architectures biased towards learning arithmetic operations (Mistry et al., 2022). This work focuses on modelling division using NALMs. The weights of NALMs are interpretable such that a discrete value represents a specific operation. For example, '-1' would represent division and '0' for no selection. Trask et al. (2018) developed the Neural Arithmetic Logic Unit (NALU), the first NALM, which can model all four arithmetic operations. The NALU consists of two sub-units; one to model addition and subtraction and another to model multiplication and division. For each input a corresponding weight value is learnt to represent the exact operation and a gate value is learnt to select between sub-units. However, multiple studies show the NALU is unstable in learning division (Schlör et al., 2020; Heim et al., 2020). In particular, their gating method responsible for selecting an operation cannot learn consistently (Madsen & Johansen, 2020). To improve the NALU, Schlör et al. (2020) developed the iNALU which applies weight and gradient clipping, sign retrieval, regularisation, reinitialisation and separating shared parameters. Even with these modifications, they find consistently learning division to a high precision to remain unattainable. Madsen & Johansen (2020) create the Neural Multiplication Unit (NMU) which only models multiplication but has significant performance gains compared to the NALU. Focusing on multiplication and division, Heim et al. (2020) developed a module which learns in the real and complex parameter space. Their results showed their Real NPU to outperform the iNALU for division. Until now, the Real NPU only has learned division on training ranges of either $\mathcal{U}[0.1,2]$ or Sobol(0,0.5) (Heim et al., 2020). It remains unclear if this module is robust to other training ranges even as a stand-alone unit. Robustness to training ranges is important as these module's applicational use comes from being part of larger end-to-end networks, where the input range into the module cannot be controlled.

## 3 Architectures

This section introduces the architectures for the (Real) NPU, NRU, and the NMRU. The (Real) NPU is an existing module, which we improve in Section 5. The NRU and NMRU are novel contributions which extend the existing NMU (see Appendix B) to do division. Since these architectures are NALMs they can be viewed as regression modules trained via supervised learning. The inputs have some underlying mathematical relation to the outputs which is modelled using division. An input is represented as a vector of features, where only certain input features are relevant to the output.

### 3.1 Real Neural Power Unit

Heim et al. (2020) develop a module to multiply and divide using the intuition from Trask et al. (2018) that *multiplicative operations are additive operations in log space*. For example, $a \times b = \exp(\ln(a) + \ln(b))$ and $\frac{a}{b} = \exp(\ln(a) - \ln(b))$. Heim et al. (2020) extends this idea into complex space. The NPU can be used with its complex form (Equation 1) requiring both a complex and real weight matrix ($\boldsymbol{W}^{\mathrm{IM}}, \boldsymbol{W}^{\mathrm{RE}}$ of shape $|\mathrm{Inputs}| \times |\mathrm{Outputs}|$), or only its real form the Real NPU (Equation 2). We focus on using the Real NPU over the NPU as the solution of the tasks in this paper can be captured using only real values meaning that the complex form is not required. For improved gradients, a relevance gate $\boldsymbol{r}$ (Equation 3) converts inputs close to 0 (i.e. irrelevant features) to 1 to avoid the resulting output evaluating to 0. A gating vector $\boldsymbol{g}$, learns to select relevant input elements, where gate values are clipped between [0,1] during training (Equation 5).

$$
\begin{aligned}
\mathrm{NPU} : y_o = \exp & \left( \sum_{i=1}^{I} (W_{i,o}^{\mathrm{RE}} \cdot \ln(r_i)) - \sum_{i=1}^{I} (W_{i,o}^{\mathrm{IM}} \cdot k_i) \right) \\
& \cdot \cos \left( \sum_{i=1}^{I} (W_{i,o}^{\mathrm{IM}} \cdot \ln(r_i)) + \sum_{i=1}^{I} (W_{i,o}^{\mathrm{RE}} \cdot k_i) \right),
\end{aligned}
\tag{1}
$$

$$
\mathrm{RealNPU} : y_0 = \exp \left( \sum_{i=1}^{I} (W_{i,o}^{\mathrm{RE}} \cdot \ln(r_i)) \right) \cdot \cos \left( \sum_{i=1}^{I} (W_{i,o}^{\mathrm{RE}} \cdot k_i) \right),
\tag{2}
$$

where

$$r_i = g_i \cdot (|x_i| + \epsilon) + (1 - g_i), \tag{3}$$

$$k_i = \begin{cases} 0 & x_i \geq 0 \\ \pi g_i & x_i < 0 \end{cases}, \tag{4}$$

and

$$g_i = \min(\max(g_i, 0), 1) . \tag{5}$$

Following Heim et al. (2020), a L1 penalty scaled by a factor $\beta$ is used when training. $\beta$ grows between predefined values $\beta_{start}$ to $\beta_{end}$ and increases every $\beta_{step} = 10,000$ iterations by a growth factor $\beta_{growth} = 10$.

### 3.2 Neural Reciprocal Unit

We propose the NRU, which can model multiplication and division, by extending the NMU motivated by the fact that *division is the multiplication of reciprocals*. The range which weight values can be is extended from [0,1] to [-1,1], where -1 represents applying the reciprocal on the corresponding input element. A NRU output element $z_o$ is defined as

$$\text{NRU} : z_o = \prod_{i=1}^{I} \left( \text{sign}(\mathrm{x}_i) \cdot |\mathrm{x}_i|^{W_{i,o}} \cdot |W_{i,o}| + 1 - |W_{i,o}| \right), \tag{6}$$

where $I$ is the number of inputs. Assuming weights are either 1 (multiply) or -1 (reciprocal), $|\mathrm{x}_i|^{W_{i,o}}$ will apply the operation on an input element. The absolute value is used so that the module only operates in the space of real numbers, as $x_i^{W_{i,o}}$ for a negative input $(x_i)$ when $-1 < W_{i,o} < 1$ results in a complex number. The use of absolute means the sign of the input must be reapplied. For the no-selection case $W_{i,o} = 0$, we want the input element to convert to 1 (the identity value), resulting in applying $\cdot |W_{i,o}| + 1 - |W_{i,o}|$. The derivative of the absolute function at 0 is undefined meaning the gradients of Equation 6 can contain points of discontinuity. To alleviate this issue, we approximate the absolute function using a scaled tanh (inspired by Faber & Wattenhofer (2020)). More formally,

$$|W_{i,o}| = \begin{cases} \tanh(1000 \cdot W_{i,o})^2 & \text{if training} \\ |W_{i,o}| & \text{otherwise} \end{cases} .$$

The scale factor (1000) controls how close to the absolute function the approximation is, where larger values give a more accurate approximation[3]. For clipping and regularisation, the same scheme as the Neural Addition Unit (NAU) (see Appendix B) is used which forces weight elements to converge to -1, 0 or 1.

### 3.3 Neural Multiplicative Reciprocal Unit

An alternate extension of the NMU, also motivated by *division being multiplication of reciprocals* is the NMRU (Equation 7). We concatenate the reciprocal of the input (plus a small $\epsilon$) to the input resulting in a module which only needs to learn selection. Hence, weights can be in the range [0,1].

$$\text{NMRU} : z_o = \prod_{i=1}^{2I} (W_{i,o} \cdot |\mathrm{x}_i| + 1 - W_{i,o}) \cdot \cos(\sum_{i=1}^{2I} (W_{i,o} \cdot k_i)) \text{ , where } k_i \qquad = \begin{cases} 0 & x_i \geq 0 \\ \pi & x_i < 0 \end{cases} . \tag{7}$$

The iteration over $2I$ represents the going through all inputs and their reciprocals. We calculate the magnitude and sign separately, joining the result at the end. The magnitude is calculated by passing the absolute of the concatenated input through an NMU architecture and the sign is calculated by using a cosine mechanism similar to the Real NPU. However, unlike the Real NPU only the weight matrix is required. The norm of the weight's gradients are clipped to 1 prior to being updated by the optimiser. This is done to alleviate the issue of exploding gradients caused by including the reciprocal to the inputs. For clipping and regularisation, the same scheme as the NMU (see Appendix B) is used.

---

[3]See Appendix I for tanh scaling experiments.

Table 1: Nine interpolation (train/validation) ranges with their corresponding extrapolation (test) range. Data (as floats) is drawn from a Uniform distribution with the range values as the lower and upper bounds.

| Interpolation | [-20, -10) | [-2, -1) | [-1.2, -1.1) | [-0.2, -0.1) | [-2, 2) | | [0.1, 0.2) | [1, 2) | [1.1, 1.2) | [10, 20) |
|---|---|---|---|---|---|---|---|---|---|---|
| Extrapolation | [-40, -20) | [-6, -2) | [-6.1, -1.2) | [-2, -0.2) | [[-6, -2), [2, 6)] | | [0.2, 2) | [2, 6) | [1.2, 6) | [20, 40) |

## 4   Single Layer Arithmetic Experiment Setup

We introduce the two main experiments used to evaluate modules, including: default parameters, train and test ranges, and evaluation metrics. The tasks evaluate the ability of a single module to divide two numbers from an input vector in two settings: **no redundancy** (2 inputs) and **with redundancy** (10 inputs). During training models will have discretisation regularisation applied to weights which enforces exact selection and precise application of arithmetic operations.

**Default parameters:**   A summary of all relevant parameters is found in Appendix C. All experiments use a mean squared error (MSE) loss with an Adam optimiser (Kingma & Ba, 2015) and 10,000 samples for the validation and test sets. Training uses batch sizes of 128 and the best model for evaluation is taken using early stopping on the validation set. All runs are over 25 different seeds. All inputs are required in the *no redundancy* setting, i.e., input size of 2. Training takes 50,000 iterations where each iteration consists of a different batch. The Real NPU uses a learning rate of 5e-3 with sparsity regularisation scaling during iterations 40,000 to 50,000. The NRU and NMRU use sparsity regularisation scaling during iterations 20,000 to 35,000 and a learning rate of 1 and 1e-2 respectively. In contrast, the *redundancy* setting uses an input size of 10, where 8 input values are not required for the final output. The total training iterations are extended to 100,000. The learning rates for the Real NPU, NRU and NMRU are 5e-3, 1e-3 and 1e-2 respectively. Sparsity regularisation scaling occurs during iteration 50,000 to 75,000 for all modules.

**Ranges:**   The ranges used for training, validation and testing is task dependant. For each task the train and validation sets are sampled from an interpolation range, whilst the test set is sampled from an extrapolation range. The interpolation ranges will not overlap with the extrapolation ranges meaning no data in the test set will ever have been seen during training.

**Evaluation metrics:**   We use the Madsen & Johansen (2019)'s evaluation scheme, consisting of three evaluation metrics: the success on the extrapolation dataset against a near optimal solution (*success rate*), the first iteration which the task is considered solved (*speed of convergence*), and the extent of discretisation towards the weights' inductive biases (*sparsity error*). Sparsity error is calculated by $\max_{i,o}(\min(|W_{i,o}|, 1 - |W_{i,o}|))$, measuring the weight element which is the furthest away from the acceptable discrete weights for the module. A success means the MSE of the trained model is lower than a threshold value (i.e., the MSE of a near optimal solution). We differ from Madsen & Johansen (2019) by using a fixed threshold value 1e-5 rather than a simulated MSE. We choose this precision as it can be guaranteed when working with 32-bit PyTorch Tensors. 95% confidence intervals (over the 25 seeds) are calculated from a specific family of distributions dependant on the metric. The success rate uses a Binomial distribution because trials (for a single seed) are either pass/fail situations. The convergence metric uses a Gamma distribution and sparsity error uses a Beta distribution. Both Beta and Gamma can easily approximate the normal distribution and support its corresponding metric.

## 5   Improving the Real NPU's Robustness

We first improve the robustness of the Real NPU against different training ranges. We use the Single Module Task with no redundancy (see Section 4) to investigate the following: (1) Is L1 regularisation required, and if so, do the regularisation parameters require tuning? (2) Does clipping the learnable parameters aid learning? (3) Does enforcing discretisation on parameters improve convergence? (4) Can the weight matrix initialisation be improved?

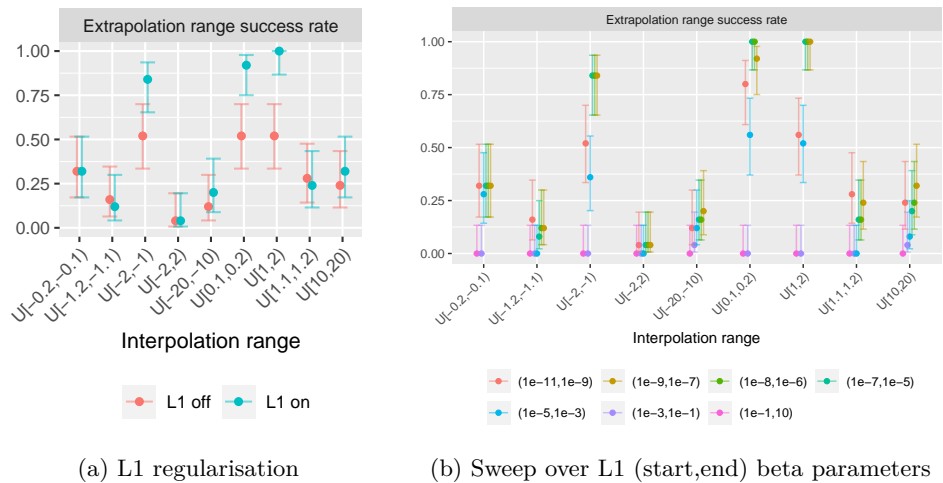

(a) L1 regularisation

(b) Sweep over L1 (start,end) beta parameters

Figure 1: Exploring the effect and sensitivity of L1 regularisation on the Real NPU

To address each question, we apply incremental modifications to the Real NPU. Modifications include an ablation study on the L1 regularisation (including a sweep over the scaling range hyperparameters), clipping, enforcing discretisation, and a more restrictive initialisation scheme. We assume that we are optimising the Real NPU to perform multiplication or division. Therefore, we trade-off the flexibility of having non-discretised weights, which enables the success of modelling the SIR data in Heim et al. (2020, Section 4.1), in favour of sparse models with discrete weight values. All the modifications can also be generalised for the NPU architecture. The ranges (Table 1) are influenced by the ranges from Madsen & Johansen (2020) as they provide good coverage.

**Is L1 regularisation required? (Yes.)** L1 encourages sparsity (i.e., zero weights) in solutions. Zero-valued weights means not to select an input and return the identity value 1. For the task, the optimal weight values require selecting all inputs and therefore non-zero values, suggesting the application of L1 could be damaging. Therefore, we compare against a model which does not use L1 regularisation, shown in Figure 1a. Removing L1 proves to be detrimental in five of the nine cases shown and only shows minor improvements in two of the nine ranges (i.e., $\mathcal{U}$[-1.2,-1.1] and $\mathcal{U}$[1.1,1.2]). Hence, we keep L1 regularisation.[4] The L1 regularisation scaling (see Section 3.1), requires setting the hyperparameters for the start ($\beta_{start}$) and end ($\beta_{end}$) scaling values. We run a sweep over six different start and end values, denoted (<start>, <end>), displaying results in Figure 1b. We find the configuration (1e-9, 1e-7) is the most successful when considering performance on all the ranges, and larger scaling values perform worse.

**Does clipping the learnable parameters help? (Yes.)** Division and multiplication are represented by weight values of -1 and 1 respectively. The current architecture does not constrain the weights which can result in large weight values. The gate weights do get clipped and saved to another variable during the forward pass, meaning after an update step the gate values can also be out of the range [-1,1]. Hence, we investigate applying clipping directly to the weight and gate values after every optimisation step. Figure 2a shows clipping is beneficial, with clipping on both weight and gate (or just on the weights) to improve over the baseline on all ranges (excluding $\mathcal{U}$[1,2] where the baseline has already achieved full success).

**Does enforcing discretisation help? (Yes.)** Modelling division in a generalisable manner requires all learnable parameters to be discrete i.e., a value from {-1, 0, 1}. Using Madsen & Johansen (2020)'s regularisation scaling scheme (see Appendix B), we penalise weights for not being discrete. We modify the scaling factor to be $\hat{\lambda} = 1$ and the regularisation to go from 'off' to 'on' between iterations 40,000 to 50,000. Figure 2b shows discretising the gate improves over the baseline but also discretising the weights

---

[4]We also experimented with using L2 regularisation but found L1 to perform better (see Appendix E).

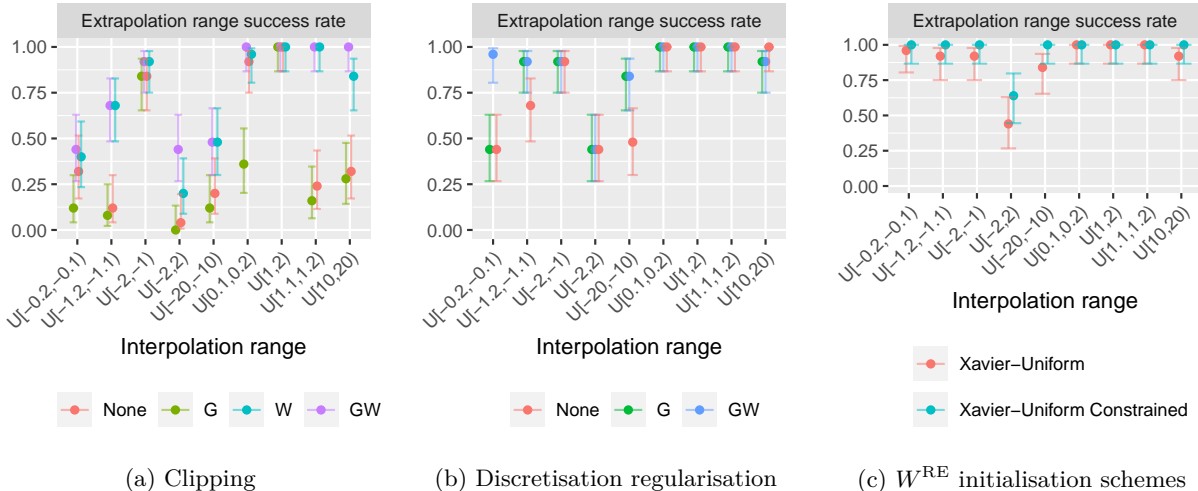

(a) Clipping      (b) Discretisation regularisation      (c) $W^{\mathrm{RE}}$ initialisation schemes

Figure 2: Effect of clipping, discretisation, and the NAU initialisation scheme on the Real NPU.

is additionally beneficial (especially for range $\mathcal{U}$[-0.2,-0.1)). $\mathcal{U}$[10,20) is the only range where the baseline outperforms using discretisation, succeeding on two additional seeds.

**Does using a more constrained initialisation help? (Yes.)** $\boldsymbol{W}^{\mathrm{RE}}$ uses a Xavier-Uniform initialisation (Glorot & Bengio, 2010), meaning weights can be initialised out of the range [-1,1]. Therefore, we use the initialisation for the Neural Addition Unit which is a constrained form of the Xavier-Uniform that does not allow the fan values of the uniform distribution to go beyond 0.5, meaning that no weight value will be out of the range [-1,1] (Madsen & Johansen, 2020). Figure 2c shows using the constrained initialisation provides improvements. For this Real NPU configuration a learning rate of 5e-3 works best (see Appendix E).

## 6 Uniform Range Datasets

We now compare learning Table 1's Uniform ranges on all modules including the NRU and NMRU for the no redundancy and redundancy setups. On the no redundancy setup (Figure 3) the NRU and NMRU achieve full success while solving the problem consistently fast and with low sparsity error, while the baseline Real NPU without modifications struggles with success on all ranges and with sparsity on the larger ranges. Applying the Real NPU modifications described in Section 5 deals with the sparsity issue and improves the robustness such that only range $\mathcal{U}$[-2,2) struggles (with a success rate of 0.64).

Introducing redundancy (Figure 4) causes failure modes to arise on all modules. The baseline Real NPU produces high sparsity errors relative to the other modules suggesting struggle with discretisation. The modified Real NPU improves over all ranges of the baseline (which were not already at full success) in terms of success, speed and sparsity (except for the sparsity in $\mathcal{U}$[10,20)). To ensure that complex weights do not fix the issue, we test the NPU module with all the modifications used on the real weight matrix but find no significant improvements (see Appendix F). The redundancy affects the NRU the most, resulting in full failures on all the negative ranges. The NMRU is the only module with success on range $\mathcal{U}$[-2,2) due to its sign mechanism (see Appendix G). It performs well over all ranges but can be outperformed by the modified Real NPU for negative ranges.

## 7 Mixed-Sign Input Datasets

The Uniform ranges results showed that the Real NPU (modified) and NRU have difficulty in learning when inputs can consist of arbitrary signed values (e.g. all positives, all negatives, or a mixture of positive and negative values) such as $\mathcal{U}$[-2,2). *We question if the failure is due to the input samples in a batch having different signs from each other, or if the problem is due to the fact data samples can be close to 0 (leading*

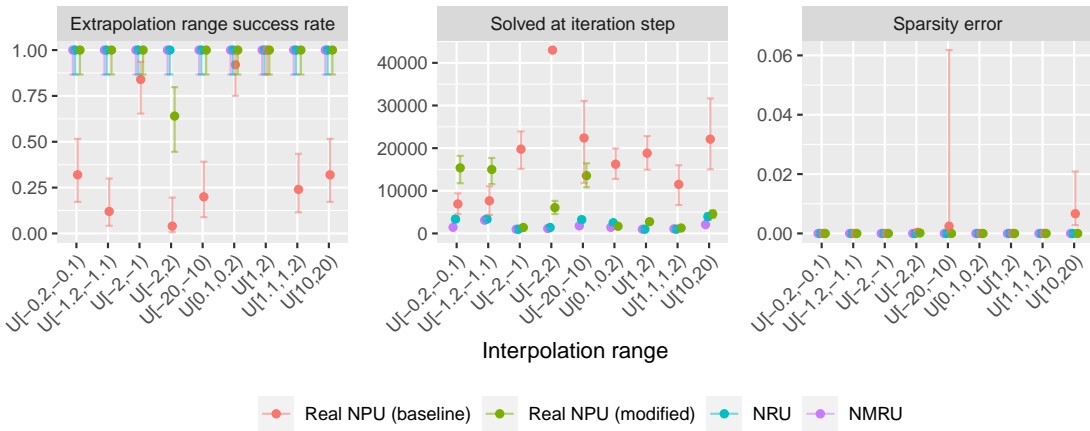

Figure 3: Division without redundancy (input size 2) on Uniform ranges.

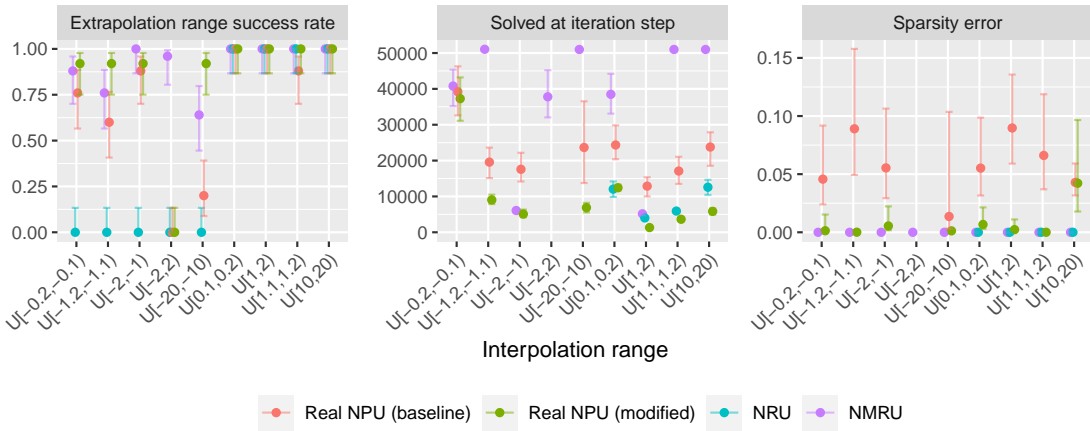

Figure 4: Division with redundancy (input size 10) on Uniform ranges.

*to singularity issues).* Five mixed-sign datasets which can control the range for each element in the input are used. The interpolation and extrapolation ranges can be found in Appendix C. Datasets 1, 2, 4 and 5 sample a positive value for one input element and a negative value for the other element. Dataset 3 samples the signs randomly. Datasets 2 and 5 avoid sampling close to 0 values to mitigate the singularity issue.

Figure 5 shows the Real NPU struggles on all these ranges while the NMU and NMRU do not. This implies that the core issue for the Real NPU is not from different input samples having different signs or due to input values being close to 0. The underlying issue is most likely correlated to each element in an input having different signs. When the denominator of the output is positive (dataset 1 or 2), the solution is found faster than when the denominator is a negative value (dataset 4 or 5). When the signs for an input element are controlled, discretisation is no problem, in contrast when the signs are arbitrary the sparsity error are slightly (though not significantly) higher. Learning with input redundancy (Figure 6), causes the Real NPU and NRU to swap in performance. The Real NPU performs significantly better than the no redundancy task on all ranges except $\mathcal{U}$[-2,2], while the NRU no longer works on any range.

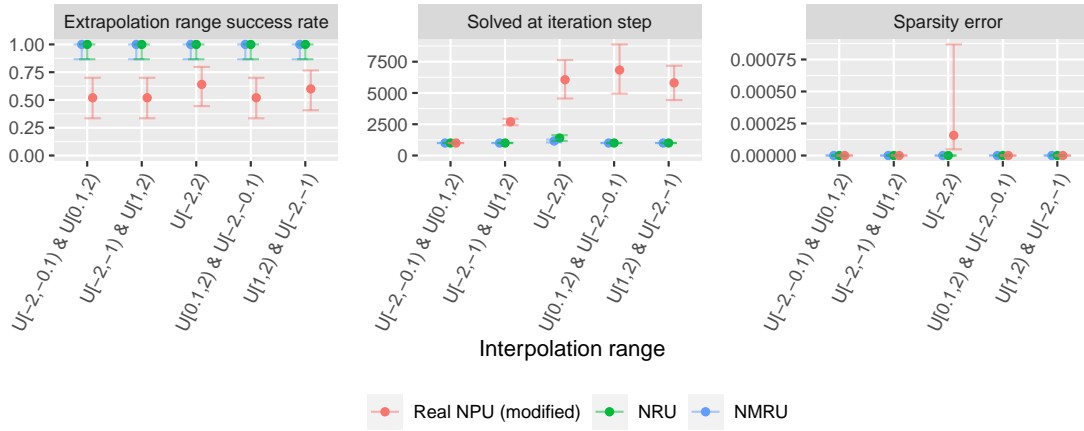

Figure 5: Division without redundancy on the mixed-sign datasets that control the sign of the input elements. The ranges are in order of the datasets (i.e. dataset 1 to 5).

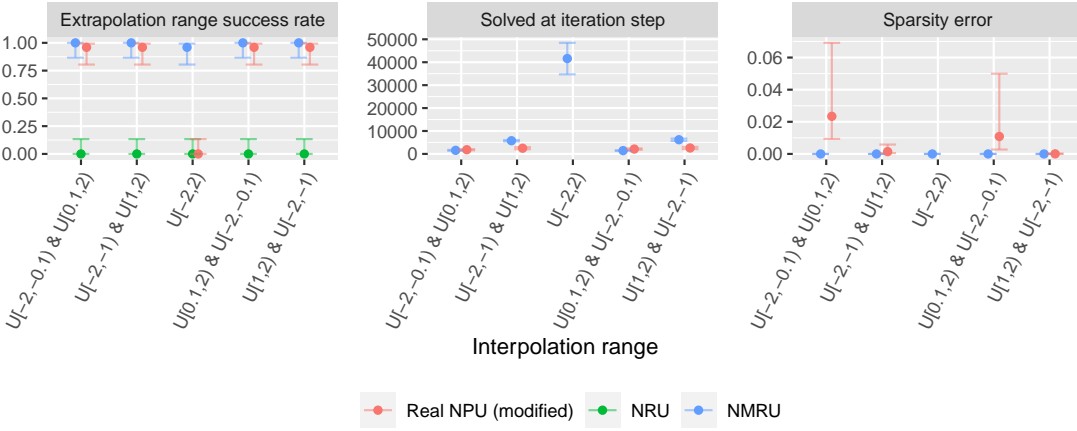

Figure 6: Division with redundancy on the mixed-sign datasets that control the sign of the input elements. The ranges are in order of the datasets (i.e. dataset 1 to 5).

## 8 More Challenging Distributions: Larger Magnitudes and Mixed-Signs

To further stress test the modules, we explore the effect of larger Uniform ranges and different distributions (i.e., Benford and Truncated Normal). The ranges are found in Table 2[5]. The Uniform ranges tests how a mixed-sign dataset is influenced by larger ranges (with magnitudes of 50 and 100). The Benford distribution also tests learning on large magnitude values. It follows a more natural distribution compared to the Uniform, known to underlie real world data such as accounting data (Hill, 1995). The Truncated Normal (TN) distributions also investigate mixed-sign datasets. A Normal distribution allows to set biases via the mean value (and is set to either -1, 0 or 1), while the truncation allows the extrapolation range not to overlap with the interpolation range. Results for the 2-input and 10-input setting are shown in Figures 7 and 8.

**Uniform distributions:** Larger ranges are found to be challenging when redundant inputs exist. On the 2-input setup, both NRU and NMRU have full success, while the Real NPU (modified) has failure cases for both Uniform distributions (with success rates of 0.72 on $\mathcal{U}$[-100,100) and 0.76 on $\mathcal{U}$[-50,50)). On the 10-input size setup, all modules fail for all runs for both ranges.

---

[5]Additional discussion and experiments for learning on small numbers is found in Appendix K

Table 2: Interpolation (train/validation) and extrapolation (test) ranges for different distributions. Data is drawn with the lower and upper bound ranges. TN = Truncated Normal in the form TN(mean, sd)[lower bound, upper bound). B = Benford. U = Uniform.

| | | | |
|---|---|---|---|
| **Interpolation** | TN(-1, 3)[-5, 10) | TN(0,1)[-5, 5) | TN(1, 3)[-10, 5) |
| **Extrapolation** | TN(-10, 3)[-15, -5) | TN(10,1)[5, 15) | TN(10, 3)[5, 15) |
| **Interpolation** | B[10, 100) | U[-100, 100) | U[-50, 50) |
| **Extrapolation** | B[100, 1000) | U[ -200, -100) ∪ [100, 200)] | U[[-100, -50) ∪ [50, 100)] |

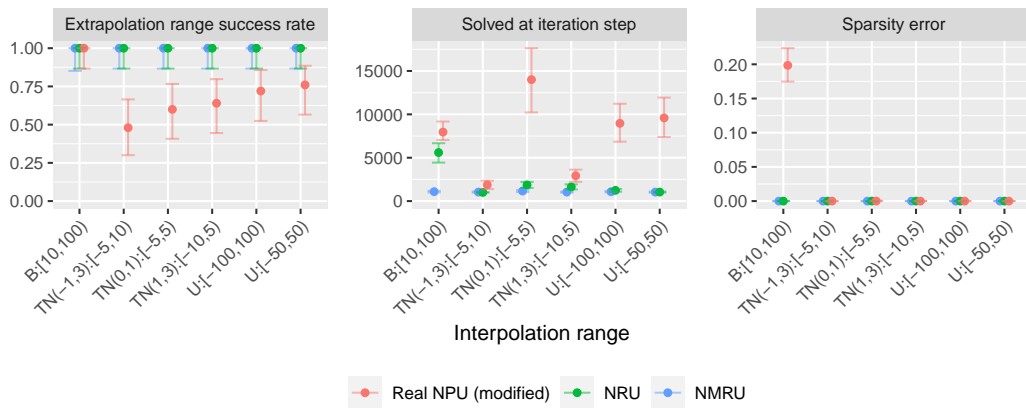

Figure 7: Division without redundancy on the Benford, Truncated Normal and Uniform distribution.

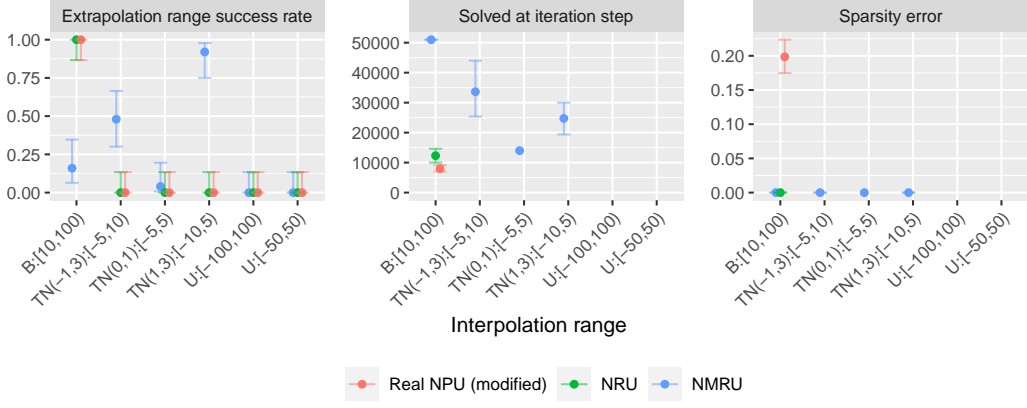

Figure 8: Division with redundancy on the Benford, Truncated Normal and Uniform distribution.

**Benford distribution:** On the 10-input setting, the NRU and modified Real NPU have full success implying the Uniform distributions failures are due to using mixed-signed inputs rather than the large ranges. The NMRU shows majority failures (failure rate 0.84) suggesting that large ranges are also an area of struggle.

**Truncated Normal distributions:** On the 2-input setup, both NRU and NMRU have full success but the Real NPU (modified) has failure cases for all three distributions (with success rates 0.48, 0.6, 0.64 respectively). When trained using the 10-input setup, both the NRU and Real NPU (modified) have no success. The NMRU's success rate greatly varies depending on the range (being 0.48, 0.04 and 0.92 for TN(-1, 3)[-5, 10), TN(0,1)[-5, 5) and TN(1, 3)[-10, 5) respectively). This suggests the NMRU works better when a majority of the inputs are likely to have the same sign and struggles with values around zero.

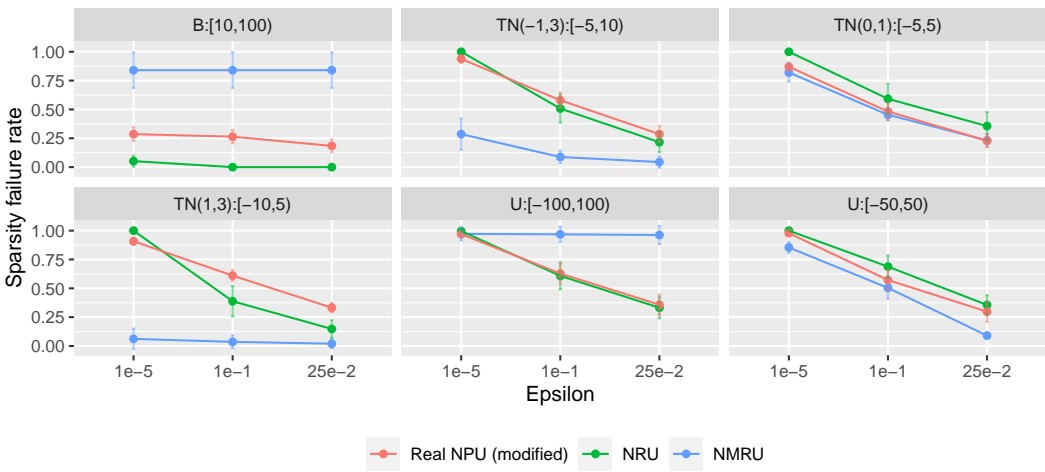

Figure 9: The sparsity failure rates (with 95% confidence intervals) for the division with redundancy task on the Benford, Truncated Normal and Uniform distributions.

## 8.1 Sparsity Failure Rates

For more insight as to how the failures are occurring, Figure 9 shows the rate of non-discrete parameters which lie more than a given epsilon away from the nearest discrete value (-1, 0 or 1) for the redundancy setup. The rates are calculated for model's best validation epoch (because of early stopping) and are averaged over all seeds and not just the successful seeds like in Figure 8. The high sparsity failure rates at the most forgiving epsilon threshold (0.25) suggests that there was high uncertainty in the learnt parameters resulting in them being unable to discretise. This is evident for the NMRU on the Benford and largest Uniform distributions. Distributions with high extrapolation success rates in Figure 8 is reflected by the low failure rate on all the epsilons. In contrast, a high reduction in failure over the epsilons occurs, then it suggests the discretisation is working but the values which parameters were converging to we not correct.

## 9 Traits of Modules when Learning on the Redundancy Setting

**Gradient difficulties with the NRU.** For insight to why the NRU performs poorly with input redundancy, we look at the gradients with respect to the weights. The partial derivative for the weights is,

$$\frac{\partial \hat{y}}{\partial w_i} = \tanh(1000w_i)(\text{sign}(x_i)|x_i|(\tanh(1000w_i)\log(|x_i|)+$$
$$2000\,\text{sech}(1000w_i)^2) - 2000\,\text{sech}(1000w_i)^2) \times \text{NRU}_{\tilde{\mathbf{x}}\in\mathbf{x}\setminus\{\mathbf{x_i}\}}(\tilde{\mathbf{x}}). \tag{8}$$

The $\text{NRU}_{\tilde{\mathbf{x}}\in\mathbf{x}\setminus\{\mathbf{x_i}\}}(\tilde{\mathbf{x}})$ term applies the NRU to all inputs excluding $x_i$ influencing the gradient values between subsequent update steps. Factoring out this term, the following observations are made: If $x_i \approx 0$ and $w_i \approx 0$ then gradients become increasingly large; if $x_i \approx 0$ and $-1 \leq w_i < 0$ then as $w_i \rightarrow -1$ all gradients for $x_i$ where $|x_i| >> 1$ become increasingly small; the gradients for $x_i = -1$ and $x_i = 1$ are 0 regardless of the value of $w_i$; if $w_i = 0$ then the gradient is 0 for all $x_i$, a result of using the tanh approximation; and, even if the sign and magnitude are calculated separately and then combined (see Appendix J) to try to control the gradient better, the problem remains. Therefore, we conclude that extending the NMU to divide using a weight of -1 is a poor choice when there are redundant inputs.

**The Real NPU's and NMRU's exploitation of multiplicative rules.** In the redundancy setting, modules with extrapolative solutions learn to exploit rules for multiplication. The NMRU exploits the inverse rule of division i.e., $a_i \cdot \frac{1}{a_i} = 1$. Since the module's input contains the reciprocals numerous extrapolative solutions exist, however this comes at the cost of finding a 'simple' solution containing non-zero weights only for relevant inputs. The Real NPU exploits the rules $a_i \cdot 0 = 0$ and $1^{a_i} = 1$ enabling non-zero weights if the

Table 3: Test accuracies of the output label for the MNIST task. The predictions and targets are rounded to 5 d.p. before the accuracy is calculated. The mean accuracy over 10-folds is given with the standard error.

|  | DIV | MLP | Real NPU (mod.) | NRU | NMRU |
|---|---|---|---|---|---|
| Test Acc. (5 d.p.) | **97.497±0.183** | 0.004±0.004 | 97.147±0.242 | 94.215±3.627 | 44.69±13.841 |

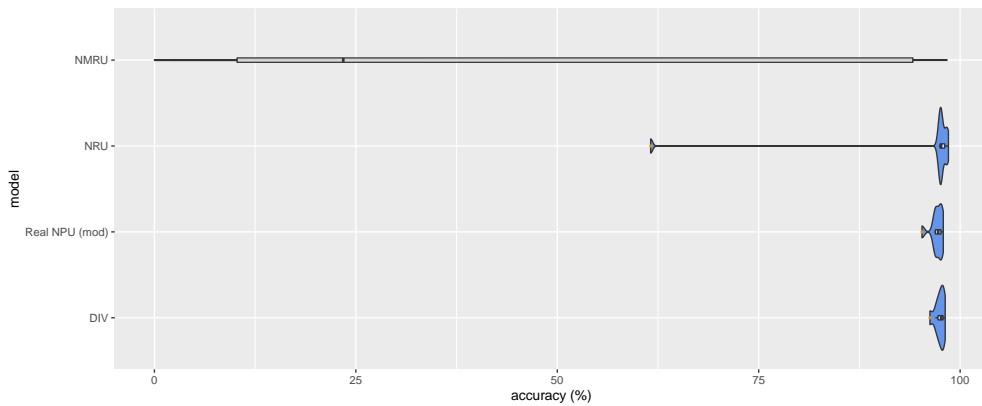

Figure 10: Test accuracies on the 5 d.p. output values.

corresponding gate value is 0. However, this can be avoided by allowing 0 to not be penalised during sparsity regularisation stage (see Appendix F); this alleviates the exploitation issue with no cost to performance.

## 10 MNIST Arithmetic

To determine if NALMs can learn in larger end-to-end networks we investigate learning to divide the labels of an image composed of two MNIST digits. Summaries of parameters can be found in Appendix D.

### 10.1 Setup and Network Architecture

Following Bloice et al. (2021), the dataset contains permutation pairs of MNIST digits side-by-side with the target label being the product of the digits, e.g. input ▣▣ with output $4(=4 \div 1)$. Importantly, although there is no overlap between the permutation pairs in the train and test set, all individual digits (between 1-9) are seen during training. E.g., the pair '54' would exist in the test set and not the train set but the digits '5' and '4' would exist in other pairs of the train set such as '15' or '47'. All instances of zero are removed from the datasets to avoid a division by zero case from occurring. The network learns a map from the input image to the labels of the two digits (digit classifier), followed by a map from the two labels to their divided value (division layer). The digit classifier is a convolutional network[6]. We separate the two digits to single digits, classify per digit and the recombine the two labels. There are three possibilities for the division layer: (1) a solved division baseline model (DIV), (2) a MLP made of 2 hidden layers with 256 hidden units and ReLU activations and L2 regularisation, and (3) a NALM being either the Real NPU (modified), NRU, or NMRU. As the DIV baseline only requires learning to classify the images to their respective labels, it is considered a strong baseline. A NALM should perform similar to the DIV baseline; if a NALM outperforms the baseline it implies the NALM can also aid learning of downstream layers aswell as learn division.

### 10.2 Metrics and Results

The output accuracy is given based on the predicted and target values rounded to 5 decimal places (d.p.) to avoid issues caused by floating point rounding. Results are taken over a 10-fold cross validation setting

---

[6]Taken from the PyTorch MNIST example `https://github.com/pytorch/examples/blob/master/mnist/main.py`

Table 4: Summary of the types division tasks the models can/cannot solve. Using redundancy means there are irrelevant inputs (10-input setup). The values are the mean success rate (out of 1) for the specific input task, bold values are the best model for the respective row.

| Redun-dancy? | Input type | Distribution | Real NPU (modified) | NRU | NMRU | Figure |
|---|---|---|---|---|---|---|
| No | Mixed-signs | Uniform | 0.56 | **1** | **1** | 5 |
| | Mixed-signs | Truncated Normal | 0.57 | **1** | **1** | 7 |
| | Negative | Uniform | **1** | **1** | **1** | 3 |
| | Positive | Uniform | **1** | **1** | **1** | 3 |
| | Large magnitude | Uniform | 0.74 | **1** | **1** | 7 |
| | Large magnitude | Benford | **1** | **1** | **1** | 7 |
| | Close to 0 | Truncated Normal | 0.6 | **1** | **1** | 7; see TN(0,1)[-5, 5] |
| | Close to 0 | Uniform | 0 | 0.17 | **0.4** | 24 |
| Yes | Mixed-signs | Uniform | 0.77 | 0 | **0.99** | 6 |
| | Mixed-signs | Truncated Normal | 0 | 0 | **0.48** | 8 |
| | Negative | Uniform | **0.92** | 0 | 0.82 | 4 |
| | Positive | Uniform | **1** | **1** | **1** | 4 |
| | Large magnitude | Uniform | 0 | 0 | 0 | 8 |
| | Large magnitude | Benford | **1** | **1** | 0.16 | 8 |
| | Close to 0 | Truncated Normal | 0 | 0 | **0.04** | 8; see TN(0,1)[-5, 5] |

and the NALM's initialisation is the same for each fold. Table 3 and Figure 10 displays the results. The MLP is not used in the violin plot so the distributions of the other modules can be better seen. The DIV baseline performs the best as expected since only the classification network requires to be learnt. The Real NPU (modified) has consistent accuracy on par with the DIV results. The NRU is less robust than the Real NPU (modified) but the better fold can outperform even the DIV. The NMRU performs the worst out of all NALMs struggling with robustness and the MLP is the worst division layer showing nearly no success across all seeds. More extensive experiments are subject to future work.

## 11 Discussion

**Single layer division robustness.** We summarise the key challenges for learning independent modules in Table 4 and give the ranges used to generate the values in Table 10. In the no redundancy setting (2-inputs), the Real NPU is challenged when the training data consists of mixed-signed inputs even with our applied improvements. Increasing the difficulty to have an input redundancy (with 8 redundant and 2 relevant input values) improves performance when the Uniform distribution is used but magnifies the issue when ranges are samples from a Truncated Normal distribution. The NRU and NMRU have strong performance across the no redundancy tasks but show failures when redundancy is included. In particular, the NRU loses its ability to learn successfully on most of the input settings. Negative ranges also becomes an issue for the NRU, in which we conclude it is not wise to use with MSE. Alternate losses can improve certain failure cases though sometimes at the cost of performance on other ranges. See Appendix M which displays results on a correlation and scale-invariant based loss. The NMRU drops most in performance on large magnitude datasets regardless the distribution. In the redundancy setup, the NMRU's robustness comes at the cost of the simplicity of the solution due to its exploitation of the identity rule; an issue the Real NPU does not have. The Truncated Normal distribution causes the greatest learning difficulties to all the modules. Learning to divide values around zero remains challenging for all modules, even on the no redundancy setup, implying an alternate method for dealing with zero denominators should be open for exploration.

**NALM can be used as part of larger networks.** The MNIST experiment shows NALMs can act as downstream layers in a non-trivial regression experiment which requires an intermediary classification network without a direct classification loss. This is promising as it implies uses of NALMs in more complex

tasks, however two points of caution should be considered. Firstly, the results show that there is not a direct correlation between the performance of a NALM in the single layer tasks to their performance if embedded in larger networks. For example, the NRU and NMRU which outperform the Real NPU on the single layer tasks perform worse in the MNIST task. Secondly, if such units are to be utilised in larger embedded networks, we encourage performing tests in the target domain before employing NALMs in the wild. Therefore, a future direction for this work and NALMs in general includes developing more challenging experimental tasks with rigorous evaluations.

**Number of learnable parameters.** The NRU requires $I \times O$ parameters, the Real NPU requires $I(O+1)$ parameters and the NMRU requires $2I \times O$ parameters. Although the NRU has the lowest parameter count, it performs the worst when redundancy is involved. The doubling of the input dimensionality in the NMRU results in more parameters, especially if the output dimension is high. Additionally, as half the inputs of the modules require being inverted (which includes the irrelevant elements), scaling difficulties can arise.

**Two-layer learning.** Once robust modules are attainable in a single layer setting, the next step would be to question performance when learning stacked modules, e.g. learning a stacked additive and multiplicative module. Madsen & Johansen (2020, Figure 2) illustrates the troubles for multiplicative models with the capacity for division. They show how a stacked summative-multiplicative module can lead to an exploding loss when the output of the summative module is close to 0 and the multiplicative model tries to divide. We recreate their setup (in Appendix N) to produce loss surfaces for the NAU-Real NPU[7], NAU-NRU and NAU-NMRU respectively. A similar issue exists with the Real NPU and NRU which use a weight range of [-1,1], whereas the NMRU whose weight's range is limited to [0,1] does not have exploding losses.

**Exploring alternate discretisation methods.** We focus on using existing discretisation techniques for NALMs to enforce weights toward values such as -1, 0 and 1. In future work, it would also be interesting to explore the effects of other types of regularisation influenced by the works from the network compression literature such as binary/ternary networks (Hubara et al., 2016; Zhu et al., 2017).

In conclusion, division remains a challenge to learn using interpretable neural networks, even for the simplest tasks. Nevertheless, by identifying the specific areas causing difficulty (e.g., training ranges), and useful architecture properties (e.g., using a sign retrieval mechanism), we hope the community has better intuition for dealing with division and developing more robust specialist modules.

### Acknowledgments

We would like to thank the anonymous reviewers who have help improve the manuscript. B.M. is supported by the EPSRC Doctoral Training Partnership (EP/R513325/1). J.H. received funding from the EPSRC Centre for Spatial Computational Learning (EP/S030069/1). The authors acknowledge the use of the IRIDIS High-Performance Computing Facility, the ECS Alpha Cluster, and associated support services at the University of Southampton in the completion of this work.

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

# Appendix

## Table of Contents

# A  Properties of a Division Module

When building a division module, the following properties should be included:

**Ability to multiply:** Without multiplication the module is limited to expressing reciprocals.

**Interpretable weights:** A good division module should produce generalisable solutions to out-of-bounds data. Using interpretable weights to represent exact operations is one way of doing so, e.g., -1 to divide, 1 to multiply, 0 to not select. For the scope of this paper, we focus on discrete weights, however fractional weights can also be considered interpretable. For example, the Real NPU can express $\frac{1}{\sqrt{x_i}}$ using a weight value of -0.5.

**Calculating the output:** This can be decomposed into three tasks: magnitude calculation, sign calculation and input selection.

**Magnitude calculation:** Refers to calculating the output value for a calculation. This is achieved using discrete weight parameters. For example, the Real NPU and NRU use a weight value of -1 for calculating reciprocals of selected input and 1 for multiplication, while the NMRU uses 1 for selecting an input element resulting in either a multiplication or reciprocal depending on the weight's position index.

**Sign of the output:** Calculating the sign value (1/-1) of the output can occur at an element level in which the sign is calculated for each intermediary value as each input element is being processed, or at the higher input level in which the sign is calculated separately from the magnitude and then applied once the final output magnitude is calculated. The NRU uses the prior method while the Real NPU and NMRU use the latter method. If an input is 0 or considered irrelevant then the output sign will be 1. (Ablation studies on the NMRU, Figure 15, suggest the latter option which separately calculates the sign to be more beneficial).

The Real NPU and NMRU use the cosine function to calculate the final sign of the module's output neuron. Below shows the state diagram of how the sign value (i.e., the state) of the output would change depending on the inputs and relevant parameters being processed. We only consider the discrete parameters for simplicity. Both the Real NPU and NMRU use the same state diagram but have different conditions for a state transition to occur.

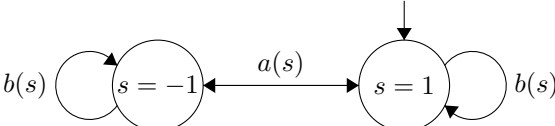

The conditions for the Real NPU transition functions $a(s) = -s$ and $b(s) = s$, where $s$ is the state value -1, or 1, are defined as follows:

$$a(s) : x_i < 0 \wedge w_{i,o} \in \{-1, 1\} \wedge g_i = 1 \ ,$$
$$b(s) : x_i \geq 0 \vee w_{i,o} = 0 \vee g_i = 0 \ .$$

Transitioning from one sign to another only occurs if the input element $(x_i)$ is negative and is considered relevant i.e. the gate $(g_i)$ and weight value $(w_{i,o})$ is non-0. In contrast, to remain at a state requires either the input element to be $\geq 0$ or not be considered relevant.

The conditions for the NMRU transition functions $a(s) = -s$ and $b(s) = s$, where $s$ is the state value -1, or 1, are defined as follows:

$$a(s) : x_i < 0 \wedge w_{i,o} = 1 \ ,$$
$$b(s) : x_i \geq 0 \vee w_{i,o} = 0 \ .$$

Transitioning from one sign to another only occurs if the input element $(x_i)$ is negative and is considered relevant i.e. the weight value $(w_{i,o})$ is 1. To remain at a state requires either the input element to be $\geq 0$ or the weight value to not select the input.

**Selection:** Not all inputs are relevant for the output value. To process any irrelevant input elements can be interpreted as converting to the identity value of multiplication/division (=1). The identity property means

that any value multiplied/divided by the identity value remains at the original number. Hence, irrelevant inputs are converted into 1 (rather than being masked out to 0). For the multiplication case, this stops the output becoming 0, and for division it avoids the divide by 0 case. For all the explored modules, a weight value of 0 will deal with the irrelevant input case. However, the Real NPU goes a step further by also having an additional gate vector with the purpose of learning to select relevant inputs. Such gating has been proven to be helpful for an NPU based module (Heim et al., 2020), but may not be necessary when dealing with weights between [0,1] like in the NRMU (see Appendix G).

## B    Neural Addition and Neural Multiplication Units' (NAU & NMU)

Madsen & Johansen (2020) develop two modules: one for dealing with addition and subtraction (the NAU) and the other for multiplication (the NMU). NAU output element $a_o$ is defined as

$$\text{NAU}: a_o = \sum_{i=1}^{I} (W_{i,o} \cdot \text{x}_i) \tag{9}$$

where $I$ is the number of inputs. The NMU output element $m_o$ is defined as

$$\text{NMU}: m_o = \prod_{i=1}^{I} (W_{i,o} \cdot \text{x}_i + 1 - W_{i,o}). \tag{10}$$

Before passing an input through a module, the weight matrix is clamped to [-1,1] for the NAU or [0,1] for the NMU. Weights are ideally discrete values, where the NAU is 0, 1, or -1, representing no selection, addition and subtraction, and the NMU is 0 or 1, representing no selection and multiplication. To enforce discretisation of weights both units have a regularisation penalty for a given period of training. The penalty is

$$\lambda \cdot \frac{1}{I \cdot O} \sum_{o=1}^{O} \sum_{i=1}^{I} \min\left(|W_{i,o}|, 1 - |W_{i,o}|\right), \tag{11}$$

where $O$ is the number of outputs and $\lambda$ is defined as

$$\lambda = \hat{\lambda} \cdot \max\left(\min\left(\frac{iteration_i - \lambda_{start}}{\lambda_{end} - \lambda_{start}}, 1\right), 0\right). \tag{12}$$

Regularisation strength is scaled by a predefined $\hat{\lambda}$. The regularisation will grow from 0 to $\hat{\lambda}$ between iterations $\lambda_{start}$ and $\lambda_{end}$, after which it plateaus and remains at $\hat{\lambda}$.

The iterations for switching on and scaling the discretisation regularisation are chosen to allow weights to have the opportunity to explore and begin to move far enough to the ideal discrete value (within 0.5), therefore the earliest regularisation is only switched on after 50% of the epochs have occurred. From our results, we have shown this works empirically. As the warmup scaling used for the regularisation occurs over many epochs, there is some leniency in the epochs to start and end the scaling without significantly impact success. In some cases, it could be possible to tune the start and end epoch for faster convergence but this was out of the scope for this study.

## C  Experiment Parameters

Tables 5 and 6 for the breakdown of parameters used in the Single Module Tasks. The parameters of Table 6 are taken from Heim et al. (2020, Section 4.1) which we confirm work empirically in Figure 1b.

Table 7 gives the interpolation and extrapolation ranges used in the mixed-sign datasets tasks.

Table 8 shows the breakdown of parameters used in the MNIST experiments. All experiments for the MNIST Tasks were trained using a single GeForce GTX 1080 GPU.

Table 5: Parameters which are applied to all modules. Parameters have been split based on the experiment. *Validation and test datasets generate one batch of samples at the start which gets used for evaluation for all iterations. † the Real NPU modules use a value of 1.

| Parameter | Without redundancy | With redundancy |
|---|---|---|
| **Layers** | 1 | 1 |
| **Input size** | 2 | 10 |
| **Total iterations** | 50,000 | 100,000 |
| **Train samples** | 128 per batch | 128 per batch |
| **Validation samples*** | 10000 | 10000 |
| **Test samples*** | 10000 | 10000 |
| **Seeds** | 25 | 25 |
| **Optimiser** | Adam (with default parameters) | Adam (with default parameters) |
| $\hat{\lambda}^{\dagger}$ | 10 | 10 |

Table 6: Parameters specific to the Real NPU modules for the Single Module Tasks.

| Parameter | Value |
|---|---|
| $(\beta_{start}, \beta_{end})$ | (1e-9,1e-7) |
| $\beta_{growth}$ | 10 |
| $\beta_{step}$ | 10000 |
| $\hat{\lambda}$ | 1 |

Table 7: Mixed-Sign Datasets: The interpolation and extrapolation ranges to sample the two input elements for a single data sample. The target expression to learn is: input 1 ÷ input 2.

| | INTERPOLATION | | EXTRAPOLATION | |
|---|---|---|---|---|
| Dataset | Input 1 | Input 2 | Input 1 | Input 2 |
| 1 | U[-2, -0.1) | U[0.1, 2) | U[-6, -2) | U[2, 6) |
| 2 | U[-2, -1) | U[1, 2) | U[-6, -2) | U[2, 6) |
| 3 | U[-2, 2) | U[-2, 2) | U[-6, -2) | U[2, 6) |
| 4 | U[0.1, 2) | U[-2, -0.1) | U[2, 6) | U[-6, -2) |
| 5 | U[1, -2) | U[-2, -1) | U[2, 6) | U[-6, -2) |

Table 8: MNIST experiment parameters.

| Parameter | Two digit MNIST |
|---|---|
| **Epochs** | 1000 |
| **Samples per permutation** | 1000 |
| **Train:Test** | 90:10 |
| **Batch Size** | 128 |
| **Train samples** | 72,000 (1 fold)/73,000 (9 folds) |
| **Test samples** | 9,000 (1 fold)/8,000 (9 folds) |
| **Folds/Seeds** | 10 |
| **Optimiser** | Adam (with default parameters) |
| **Criterion** | MSE |
| **Learning rate** | 1e-3 |
| $\lambda_{start} - \lambda_{end}$ **epochs** | 30-40 |
| $\hat{\lambda}$ | 2 |
| **grad norm clip** | MLP = None; Real NPU = None; NRU = 1; NMRU = None |

### C.1 Parameter Initialisation

We give the initialisations used on the different module parameters:

**Real NPU**: The real weight matrix uses the Pytorch's Xavier Uniform initialisation. The gate vector initialises all values to 0.5. (This is the same initialisation used in Heim et al. (2020).)

**NPU**: The imaginary weight matrix is initialised to 0. The rest of the parameters are initialised same as the Real NPU. (This is the same initialisation used in Heim et al. (2020).)

**NRU**: The weight matrix uses a Xavier Uniform initialisation which can have a maximum range between -0.5 to 0.5 (depending on the network sizes). (This is the same initialisation the Neural Addition Unit uses (Madsen & Johansen, 2020).)

**NMRU**: The weight matrix uses a Uniform initialisation which can have a maximum range between 0.25 to 0.75 (depending on the network sizes). (This is the same initialisation the Neural Multiplication unit uses (Madsen & Johansen, 2020).)

## D   Hardware and Time to Run Experiments

All experiments were trained on the CPU, as training on GPUs takes considerably longer. All Real NPU experiments were run on Iridis 5 (the University of Southampton 's supercomputer), where a compute node has 40 CPUs with 192 GB of DDR4 memory which uses dual 2.0 GHz Intel Skylake processors. All NRU and NMRU experiments were run on a 16 core CPU server with 125 GB memory 1.2 GHz processors.

Table 9 displays time taken for each experiment to run a single seed for a single range. Timings are based on a single run rather than the runtime of a script execution because the queuing time from jobs when executing scripts is not relevant to the experiment timings. For a single model, a single experiment would have 225 runs (for 9 training ranges and 25 seeds).

Table 9: Timings of experiments.

| Experiment | Model | Approximate time for completing 1 seed (mm:ss) |
|---|---|---|
| No redundancy (size 2) | Real NPU | 03:20 |
| | NRU | 02:00 |
| | NMRU | 03:00 |
| With redundancy (size 10) | Real NPU | 05:30 |
| | NRU | 05:00 |
| | NMRU | 05:15 |

## E   Real NPU; Single Module Task (without Redundancy): Additional Experiments

Figure 11 shows results of using the NPU for the 2-input task. Of the 9 tested ranges, L2 has a lower success rate than L1 for 5 ranges and has the same success rate for the remaining 4 ranges. If L2 regularisation is used instead of no regularisation, it performs worse in 3 (of the 9) ranges, better on 3 ranges and the same on the remaining 3 ranges.

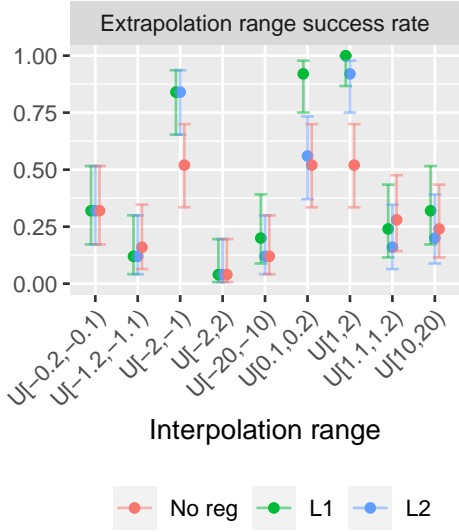

Figure 11: Applying no regularisation, L1 regularisation and L2 regularisation to enforce sparsity in weights.

Figure 12 displays the effect of different learning rates for the modified Real NPU. A learning rate of 5e-3 has the best performance over all ranges.

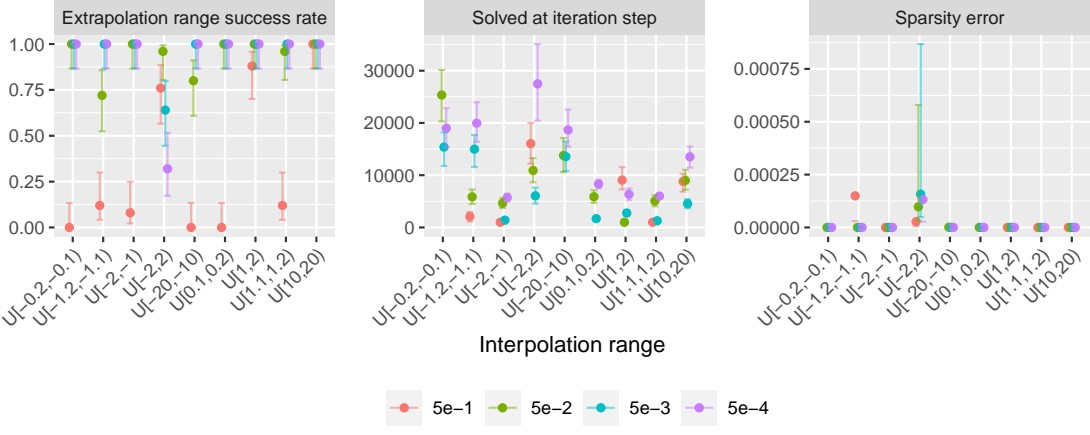

Figure 12: Different learning rates on the Real NPU (mod) for the Single Module Task (no redundancy).

## F    Real NPU; Single Module Task (with Redundancy): Additional Experiments

We test the NPU module with all the modifications used on the real weight matrix. Also, assuming the global solution only uses the real weights, we enforce the complex weights to be clipped between [-1,1] and to go to 0 during the regularisation stage using a L1 penalty. Figure 13 shows the complex weights without any constraints hinders success and convergence speeds of negative ranges. Applying clipping and regularisation constraints does not result in any significant improvements against the Real NPU results.

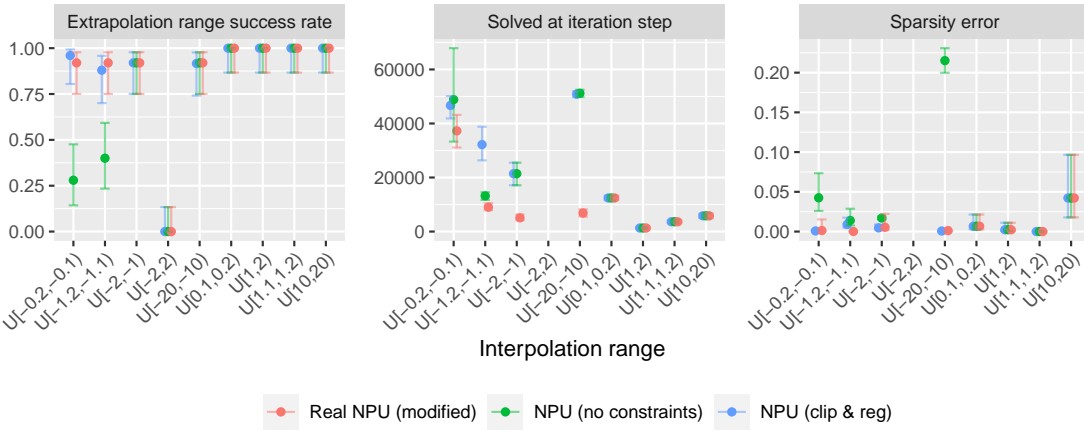

Figure 13: Adapting the Real NPU to use complex weights (NPU) on the Single Module Task with redundancy. Compares the NPU architecture with the Real NPU modifications (i.e. NPU (no constraints)) and the same model but with the imaginary weights clipped to [-1,1] and L1 sparsity regularisation on the complex weights (i.e. NPU (clip & reg)).

Figure 14 shows how modifying the Real NPU's weight discretisation to not penalise weights at 0 does not affect success.

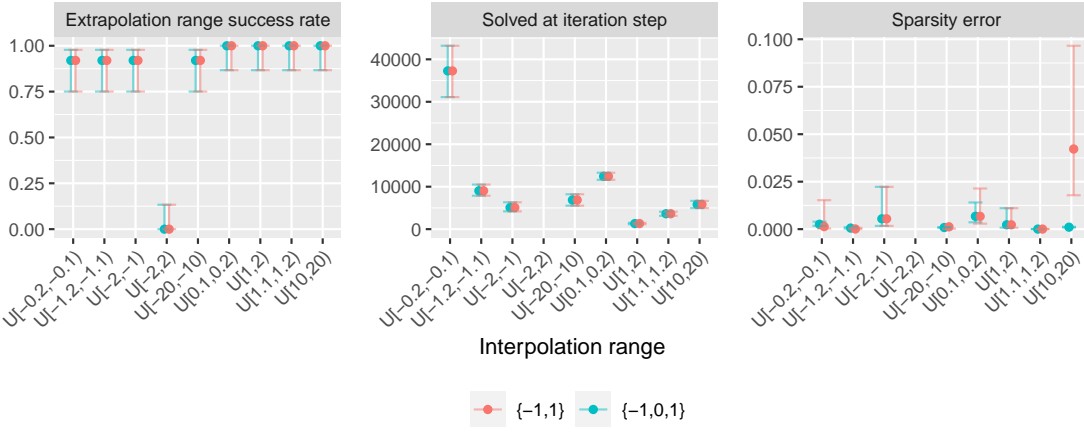

Figure 14: Comparing weight discretisation on the NPU weights which penalises not having weight of $\{-1, 1\}$ vs $\{-1, 0, 1\}$.

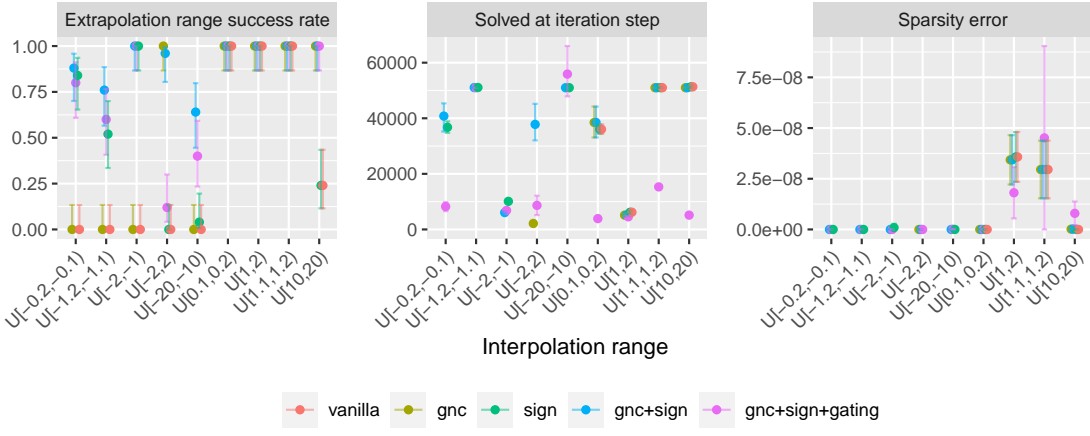

Figure 15: Ablation study for the NMRU.

## G NMRU; Single Module Task with Redundancy: Additional Experiments

This section further explores the NMRU architecture.

Figure 15 shows an ablation study on different components of the NMRU architecture. Removing both the sign retrieval and grad norm clipping performs poorly over a majority of ranges (including positive ranges). Gradient norm clipping alone is unable to solve the issue in learning negative ranges, however fully succeeds on the $\mathcal{U}$[-2,2] range. Using the sign retrieval without the gradient clipping gains successes for the negative ranges, though performance on $\mathcal{U}$[2,-2] is affected. However, including both gradient clipping and sign retrieval results in separating the calculation of the magnitude of the output and its sign while having reasonable gradients, gaining the most improvement over the vanilla NMRU. Further including a learnable gate vector (like the Real NPU), which is applied to the input vector, hinders performance. The largest solved at iteration step seems to be bounded at approximately 50,000 iterations which correlates to the point at which the sparsity regularisation begins, which highlights the importance of discretisation. Even with the different ablations, the sparsity errors of the successful seeds remain extremely low (which is not always the case for the Real NPU (see Figure 4)).

Figure 16 shows the effect of using different learning rates on the NMRU (with grad norm clipping and sign retrieval) using an Adam optimiser. Too low a learning rate struggles on the mixed-sign range $\mathcal{U}$[-2,2]. Too high a learning leads to no success on multiple ranges.

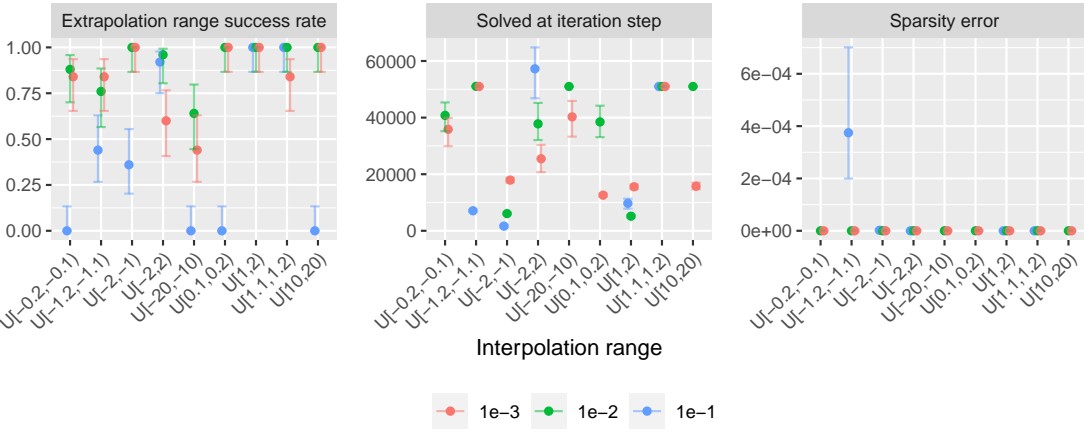

Figure 16: Effect of different learning rates on the NMRU

Figure 17 compares training the NMRU with either an Adam and SGD optimiser. As expected, Adam outperforms SGD in all ranges (except two, where both perform equally). This difference in performance can be accounted for by Adam's ability to scale the step size of each weight, which can complement the clipped gradient norm of the NMRU, in contrast to the SGD's global step size.

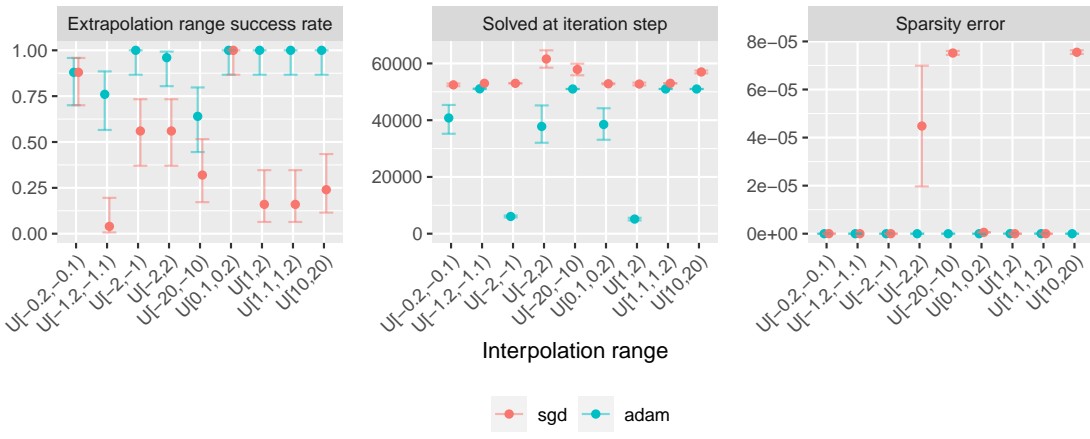

Figure 17: Effect of optimiser on the NMRU. SGD = Stochastic Gradient Descent.

## H    NRU; the Single Module Task (no Redundancy): Effect of Learning Rate

Figure 18 displays the effect of different learning rates for the NRU. A learning rate of 1 gets full success on all ranges with performance deteriorating as the learning rate reduces.

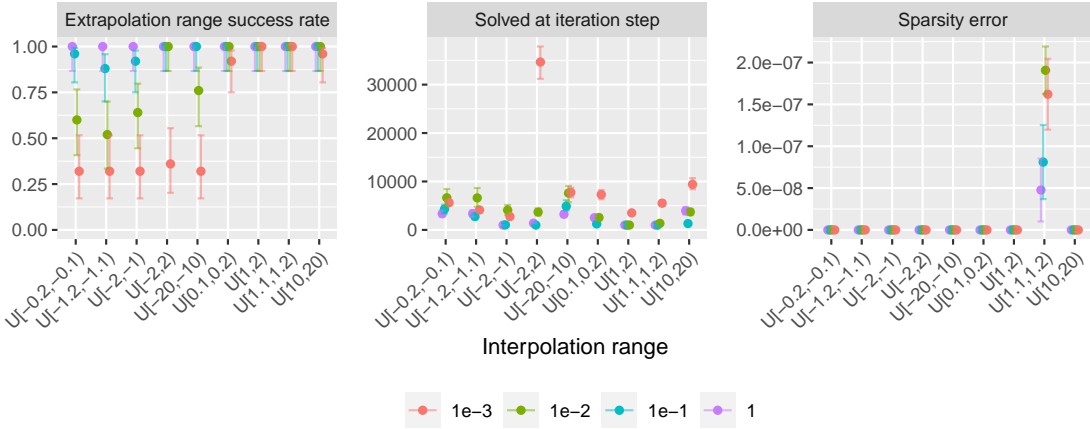

Figure 18: Different learning rates on the NRU for the Single Module Task (no redundancy)

## I  NRU; Single Module Task (no Redundancy); Tanh Scale Factor

Figure 19 shows the impact of changing the tanh scale factor. We find lager scale factors work better with a factor of 1000 being the best. This correlates to the findings in Faber & Wattenhofer (2020, Figure 5).

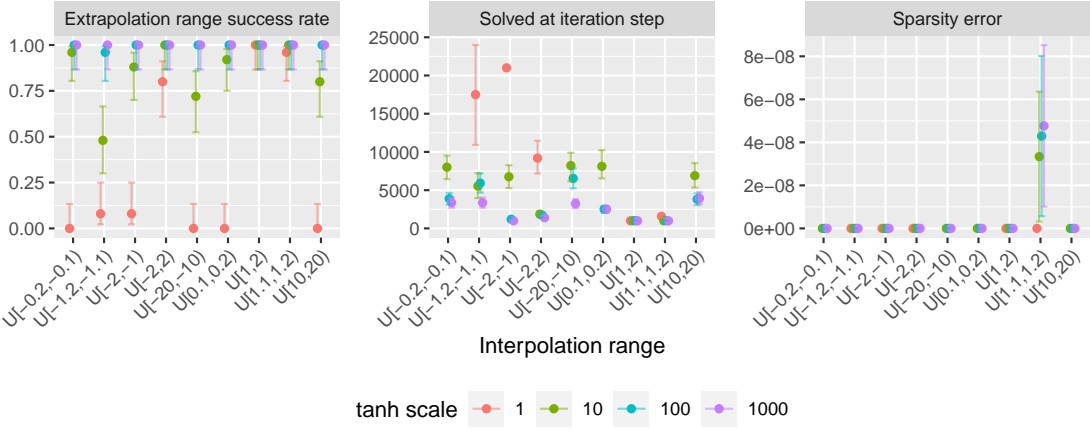

Figure 19: Effect of the tanh scale factor for the NRU on the 2-input setting.

## J  NRU; Single Module Task (with Redundancy): Calculating the Sign Separately

The 'separate NRU' module calculates the magnitude and sign separately and then combines them using multiplication together once all input elements are accounted for. The following definition is used to calculate a NRU with separate magnitude and sign calculation,

$$z_o = \prod_{i=1}^{I} \left( |\mathrm{x}_i|^{W_{i,o}} \cdot |W_{i,o}| + 1 - |W_{i,o}| \right) \cdot \prod_{i=1}^{I} \mathrm{sign}(\mathrm{x}_i)^{\mathrm{round}(W_{i,o})} \ . \tag{13}$$

Figure 20 shows results, where the separate sign method shows no difference in success to the original NRU architecture.

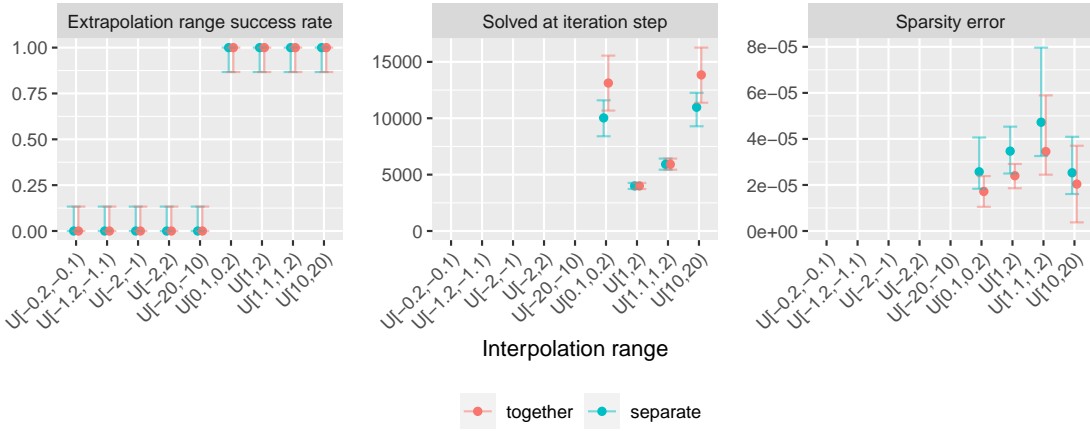

Figure 20: NRU on the redundancy experiment comparing a module which calculates the magnitude and sign together vs calculating the magnitude and sign separately and then combining them.

## K    Division by Small Values:

The discontinuous nature of division at zero results in the inability to provide a computational value for the output/gradient and causes neighbouring values to have large gradients. To understand the extent of this issue when learning, we explore learning to divide by values close to zero using three tasks with increasing difficulty: 1) learning to take the reciprocal of a single input, 2) taking the reciprocal of the first input given two inputs, and 3) dividing the first input by the second given two inputs.

### K.1    Impact of the Singularity Issue on Gold Solutions

Figure 21 plots the test error assuming the module weights are set to the 'gold' solution for the three tasks. As the range values become closer to zero, the test error thresholds become increasingly large. Therefore, even with the correct weights, relying on the test errors alone as an indicator become increasingly deceptive with values close to zero. The Real NPU has larger test errors for all tasks and ranges, caused by adding $\epsilon$ to the input (see Equation 3). Setting $\epsilon = 0$ reduces the test error at the cost of the ability to deal with zero-valued inputs. Appendix K.2 provides the corresponding experimental results finding that only modelling reciprocals can be learnt with extremely small values.

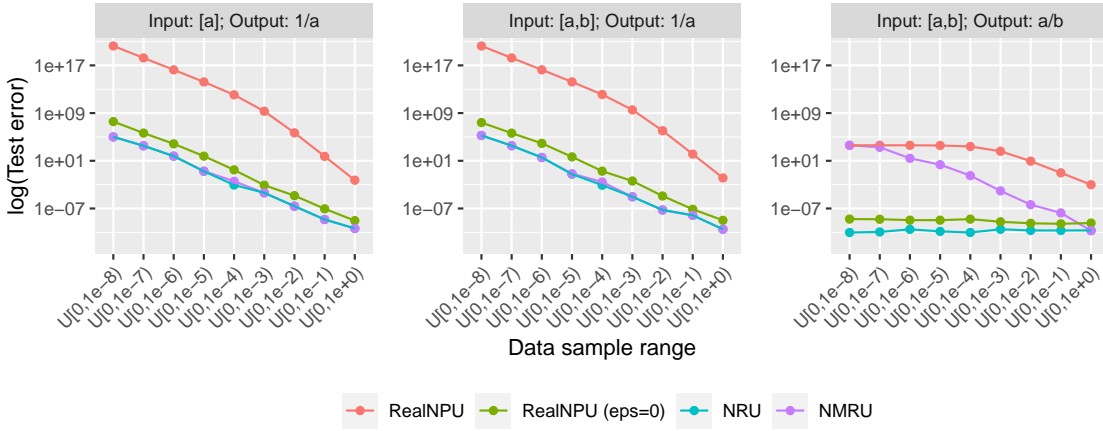

Figure 21: Effect of the singularity issue on the Real NPU, NRU and NMRU over increasing input ranges. Left: Reciprocal for an input size of 1 (no redundancy). Middle: Reciprocal for an input size of 2 (with redundancy). Right: Division for an input size of 2 (no redundancy).

### K.2    Experimental Results

This section shows the results on trying to learn the reciprocal/division of values close to zero using the Real NPU, NRU and NMRU. We train and test on the ranges where the lowest bound is 0 and the upper bounds are: 1e-4, 1e-3, 1e-2, 1e-1 and 1. Unless stated otherwise, the hyperparameters of a model are set to what is used for the Single Layer Task without redundancy. The first task runs for 5,000 iterations with no regularisation for any module. The second and third tasks both run for 50,000 iterations.

Due to precision errors, a solution with the ideal parameters will not evaluate to a MSE of 0. Therefore, we calculate thresholds which the test MSE should be within. A threshold value for a task is calculated from evaluating the MSE of each range's test dataset for each module, using the 'golden' weight values and adding an epsilon term[8] to the resulting error which takes into account precision errors. All experiments are run using 32-bit precision.

---

[8]The term is the Pytorch default eps value, torch.finfo().eps

In general, successful runs take longer to solve as the input ranges become smaller. The simplest task, of taking the reciprocal when the input size is 1 (Figure 22) is achieved with ease for all modules, though for $\mathcal{U}[0,1\text{e-}4)$, we find the NRU begins to start struggling.

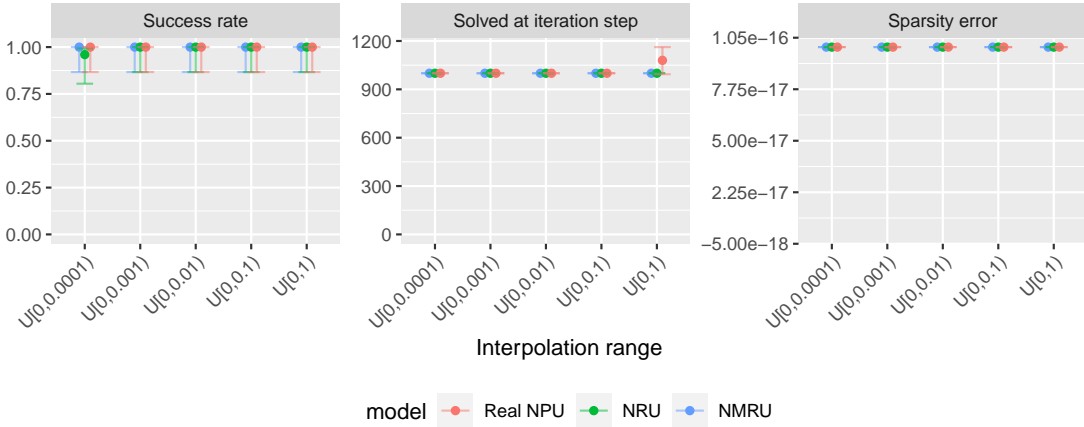

Figure 22: Input: [a], output $\frac{1}{a}$. Learns reciprocal when there is no input redundancy.

Introducing a redundant input (Figure 23) greatly impacts performance with only the NMRU able to achieve reasonable success for the larger ranges. The successes shown for the Real NPU at range $\mathcal{U}[0, 1\text{e-}4)$ are false positives caused by the $\epsilon$ in the architecture used for stability. Test false positives can also be indicated by the high sparsity error of the weights.

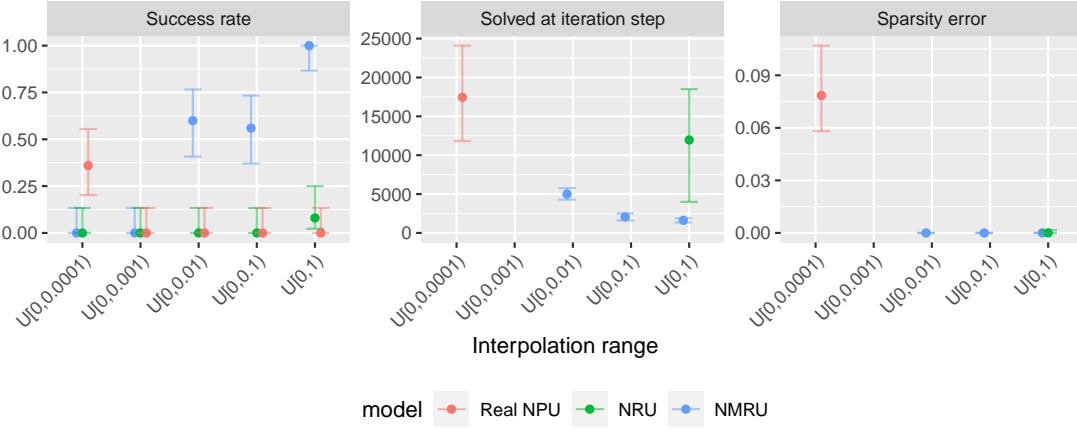

Figure 23: Input: [a,b], output $\frac{1}{a}$. Learns reciprocal of the first input when there is redundancy.

Modifying the task to division (Figure 24), meaning the redundant input is now relevant, shows improvement for the NMRU and NRU for the larger ranges.

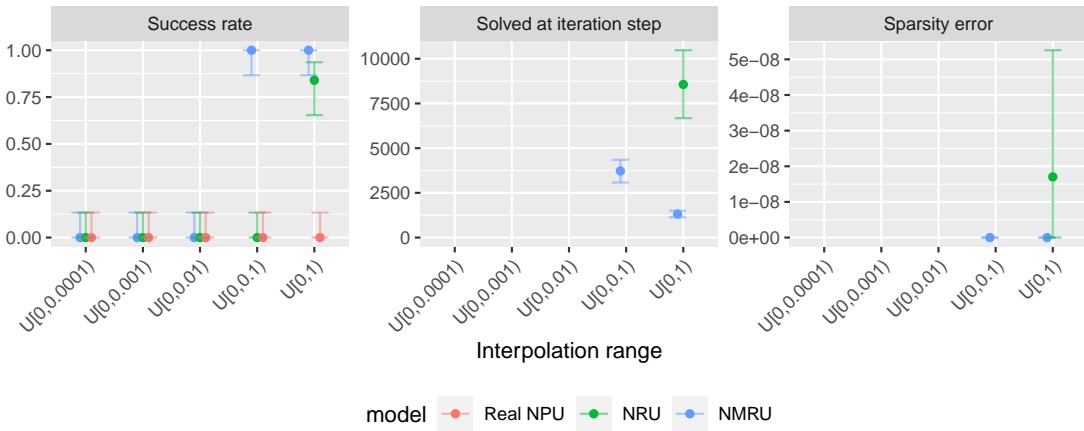

Figure 24: Input: [a,b], output $\frac{a}{b}$. Learns division of the first and second value when there is no redundancy.

## L    Ranges Used for the Single Layer Task Summary Table

Table 10 shows the ranges used to generate the summary statistics. Note that even though the interpolation ranges are given (to make it easier to compare against the relevant Figures), it is the success rate on the extrapolation range which is used in the table summary.

Table 10: The relevant ranges used to calculate the summary statistics in Table 4.

| Redun-dancy? | Input type | Distribution | Figure | Interpolation Ranges |
|---|---|---|---|---|
| No | Mixed-signs | Uniform | 5 | All 5 ranges: U[-2,-0.1) & U[0.1,2), U[-2,-1) & U[1,2), U[-2,2), U[0.1,2) & U[-2,-0.1) and U[1,2) & U[-2,-1) |
| | Mixed-signs | Truncated Normal | 7 | All 3 TN ranges: TN(-1,3): [-5,10), TN(0,1):[-5,5) and TN(1,3):[-10,5) |
| | Negative | Uniform | 3 | Only pure negative ranges: U[-0.2,-0.1), U[-1.2,-1.1), U[-2,-1) and U[-20,-10) |
| | Positive | Uniform | 3 | Only pure positive ranges: U[0.1,0.2), U[1,2), U[1.1,1.2) and U[10,20) |
| | Large magnitude | Uniform | 7 | U[-100,100) and U[-50,50) |
| | Large magnitude | Benford | 7 | B[10,100) |
| | Close to 0 | Uniform | 24 | All 5 ranges: U[0,0.0001), U[0,0.001), U[0,0.01), U[0,0.1) and U[0,1) |
| | Close to 0 | Truncated Normal | 7 | TN(0,1)[-5, 5) |
| Yes | Mixed-signs | Uniform | 6 | All 5 ranges: U[-2,-0.1) & U[0.1,2), U[-2,-1) & U[1,2), U[-2,2), U[0.1,2) & U[-2,-0.1) and U[1,2) & U[-2,-1) |
| | Mixed-signs | Truncated Normal | 8 | All 3 TN ranges: TN(-1,3): [-5,10), TN(0,1):[-5,5) and TN(1,3):[-10,5) |
| | Negative | Uniform | 4 | Only pure negative ranges: U[-0.2,-0.1), U[-1.2,-1.1), U[-2,-1) and U[-20,-10) |
| | Positive | Uniform | 4 | Only pure positive ranges: U[0.1,0.2), U[1,2), U[1.1,1.2) and U[10,20) |
| | Large magnitude | Uniform | 8 | U[-100,100) and U[-50,50) |
| | Large magnitude | Benford | 8 | B[10,100) |
| | Close to 0 | Truncated Normal | 8 | TN(0,1)[-5, 5) |

## M    Effect of Different Losses on the Single Module Task (with Redundancy)

Different losses induce different loss landscapes impacting the areas of success for a module. We explore the effects of three different losses including the MSE, Pearson's Correlation Coefficient (Equation 15), and the Mean Absolute Precision Error (Equation 16). We use the division task with 10 inputs. The properties of each loss is summarised in Table 11. All experiment parameters match the original MSE runs in the main experiments. The only difference is the loss used.

Table 11: The properties of different loss functions.

|  | MSE | PCC | MAPE |
|---|:---:|:---:|:---:|
| Batch mean | ✓ | ✓ | ✓ |
| Standardisation |  | ✓ | ✓ |
| Difference of prediction from target | ✓ |  | ✓ |
| Projection |  | ✓ |  |
| Mean centering |  | ✓ |  |

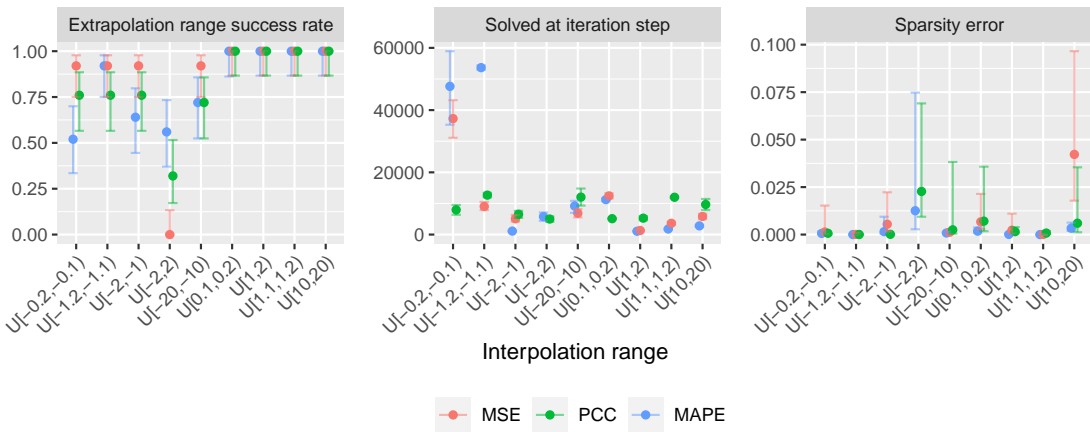

Figure 25: Single Module Task with redundancy on the Real NPU, comparing different loss functions.

$$v_{x,i} = (\hat{y}_i - \bar{\hat{y}}), \quad s_x = \sqrt{\text{clamp}(\frac{1}{N}\sum_i^N v_{x,i}^2, \, \epsilon)}$$

$$v_{y,i} = (y_i - \bar{y}), \quad s_y = \sqrt{\text{clamp}(\frac{1}{N}\sum_i^N v_{y,i}^2, \, \epsilon)} \tag{14}$$

$$r = \frac{1}{N}\sum_i^N (\frac{v_{x,i}}{s_x + \epsilon} \cdot \frac{v_{y,i}}{s_y + \epsilon})$$

$$\text{pcc loss} := 1 - r \tag{15}$$

where N is the batch size, and the means ($\bar{\hat{y}}$ and $\bar{y}$) are taken over the batch. $\epsilon$ is used to provide better numerical stability. The clamping refers to setting the minimum of the values to $\epsilon$.

$$\text{mape loss} := \frac{1}{N}\sum_i^N (\frac{|y_i - \hat{y}_i|}{y_i}) \tag{16}$$

**Real NPU (Figure 25)** Both the Real NPU and MAPE are able to get success on the $\mathcal{U}$[-2,2) range, which the MSE completely fails on, implying that having a loss with standardisation is useful. However, to gain successes in the mixed-sign range, the other negative ranges have reduced in success for both PCC and MAPE. Both speed and sparsity retain similar performance to MSE in a majority of cases, with PCC solving especially fast for all tested ranges.

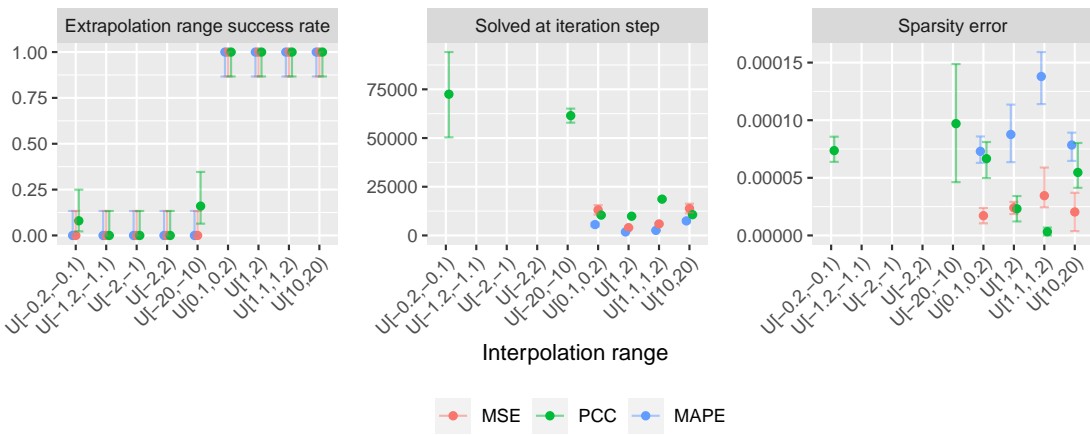

Figure 26: Single Module Task with redundancy on the NRU, comparing different loss functions.

**NRU (Figure 26)** Different losses have little effect on the NRU. All three losses perform well on the positive ranges. Compared to the Real NPU, the PCC loss on the NRU takes longer to converge to a success for negative ranges.

**NMRU (Figure 27)** All three loses perform reasonably well, with the PCC struggling the most. Unlike the other units, $\mathcal{U}$[-20,-10) causes the most trouble, whereas $\mathcal{U}$[-2,2) gains near to full success on two of the three losses.

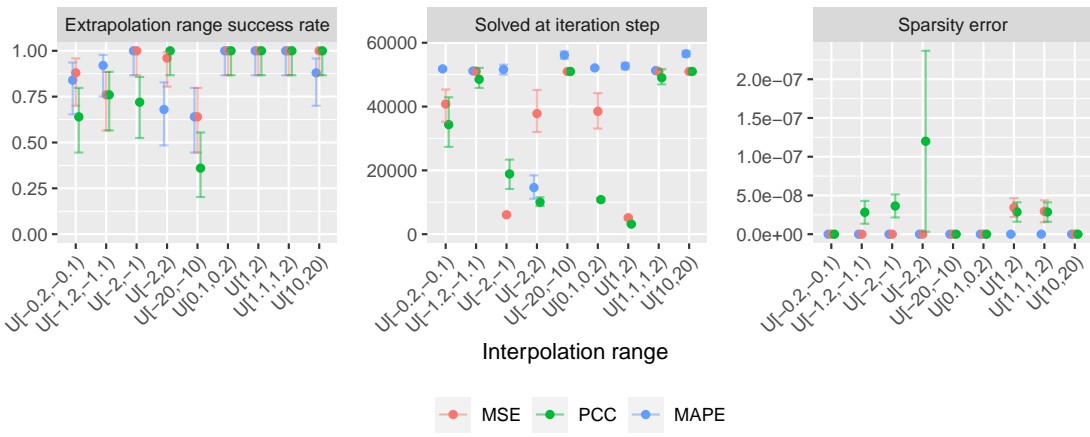

Figure 27: Single Module Task with redundancy on the NMRU, comparing different loss functions.

## N   RMSE Loss Landscapes

The following shows the Root Mean Squared loss curvature for the NAU stacked with either a RealNPU, NRU, or NMRU. "The weight matrices are constrained to $\mathbf{W}_1 = \left[\begin{smallmatrix} w_1 & w_1 & 0 & 0 \\ w_1 & w_1 & w_1 & w_1 \end{smallmatrix}\right]$, $\mathbf{W}_2 = \left[\begin{smallmatrix} w_2 & w_2 \end{smallmatrix}\right]$. The problem is $(x_1 + x_2) \cdot (x_1 + x_2 + x_3 + x_4)$ for $x = (1, 1.2, 1.8, 2)$" (Madsen & Johansen, 2020). The ideal solution is $w_1 = w_2 = 1$, though other valid solutions do exist e.g., $w_1 = -1, w_2 = 1$. The NMRU's weight matrix would be $\mathbf{W}_2 = \left[\begin{smallmatrix} w_2 & w_2 & 0 & 0 \end{smallmatrix}\right]$, and the Real NPU's $\mathbf{g} = \left[\begin{smallmatrix} 1 & 1 \end{smallmatrix}\right]$.

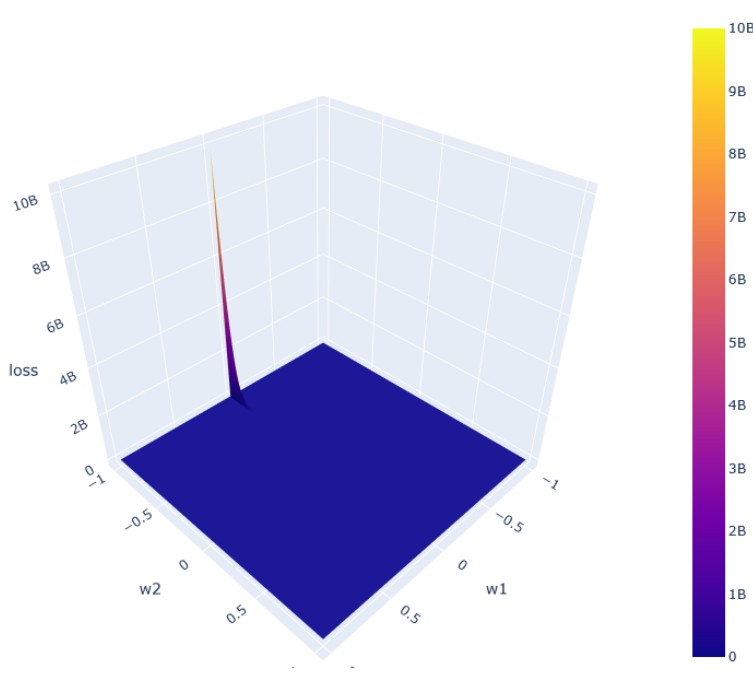

(a) NAU-Real NPU (where $\epsilon = 1e - 5$)

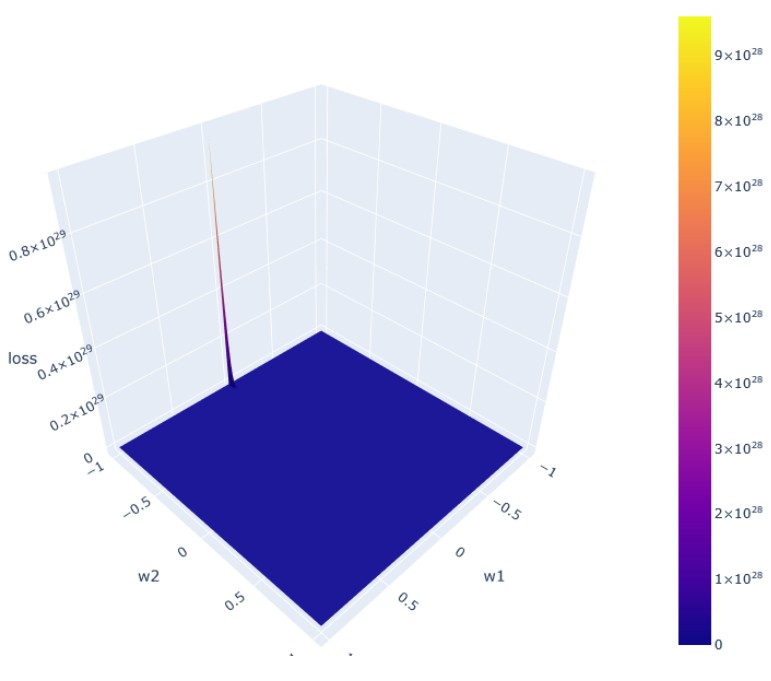

(b) NAU-NRU

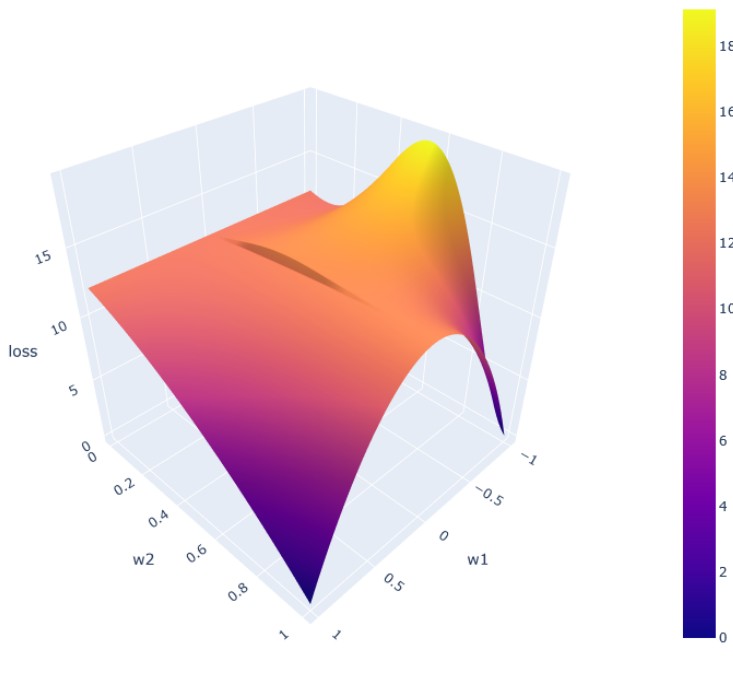

(c) NAU-NMRU

Figure 28: Enlarged loss landscapes of different stacked summative-multiplicative units.

