# OpenReview forum: "Exploring the Learning Mechanisms of Neural Division Modules"
_TMLR — Accepted by TMLR_

### Review · Reviewer_fzJX · 2022-06-18

**Summary Of Contributions:**

The paper follows a line of research on Neural ALUs (Arithmetic Logical Units). Given the crucial role that ALUs play in modern processors, the Neural ALU is supposed to be a learned equivalent that is comprised of (roughly) neural-network building blocks. Importantly, the Neural ALU needs to be able to learn arithmetic/logic computation that generalizes to arbitrary inputs (of the same numerical type). Building and learning Neural ALUs has been a challenge in the field for a few years now - the most elusive arithmetic computation being division. This is the problem that this paper aims to make progress on. It is phrased mainly as a problem of information-selection and routing (which roughly requires binary or ternary weights).

In particular, the paper makes the following main contributions:
 * Empirical investigation of 4 improvements to a previously proposed neural module to solve the problem (main idea: division via log-space subtraction). The improvements are: L1 regularization, parameter clipping, enforcing binary/ternary parameter values, non-standard initialization. These improvements are fairly obvious given the problem at hand, but nonetheless they lead to a significant increase in performance of the baseline method, as is shown by a series of well-conducted ablations.
 * Proposal of two alternative neural modules, where the main idea is to implement division via multiplication with the reciprocal. The modules require differentiability (to optimize parameters via SGD), and, like the baseline method, binary or ternary parameter values (ideally). The proposals are theoretically sound, and seem fairly straightforward. Both proposals are empirically compared against the (modified) baseline method and often perform favorably.
 * The main finding is that the unmodified baseline method can be improved, and that the new proposals perform even better in some cases. The paper also shows (sometimes systematic) limitations of all three methods, which highlights the need for further research to find a general Neural Division Module.

The main evaluation of the paper is on two types of tasks:
 * No redundancy: the module is fed with two one-dimensional inputs and needs to produce an output value corresponding to one input divided by the other.
 * Redundancy: same as above, but in addition the module is fed eight non-informative inputs that the module needs to learn to ignore.

Overall I think the paper shows a large range of modifications of the baseline method and ablations for all three modules under a large set of experimental conditions. In its current form the manuscript and set of experiments feels a bit unorganized and rushed. I think the experiments could be reorganized and trimmed down to have an easier-to-read main text without compromising the narrative and main points. There is a lot of work and results in the paper, but the focus could be sharpened (I know it can be tempting to present everything that was tried and investigated, but at least in the main paper less can be more sometimes). Having said that, I think the main ingredients for an acceptance are there but I want to encourage the authors to try and streamline and publish the best version of this paper (rather than just getting it out the way and moving on).


**Broader Impact Concerns:**

I have no concerns regarding the broader impact.

**Requested Changes:**

Major/important changes
1. Perhaps the main experiments can be distilled down into a smaller, more informative, set with the remaining ones moved to the appendix. In particular, why does Table 1 only have a single mixed-sign experiment and only a single two-sided extrapolation experiment? I would have liked to see more of the datasets that went into Figure 4 and Table 2 as the main experiment (and only one or two representative single-sign, one-sided extrapolation experiments). Though this is listed as a major change, I am OK if this is not addressed - but personally think that it would be a major improvement (and would at least like to hear the authors’ opinion on why the current presentation is advantageous).
2. Cherry-picking: were some additional experiments selected to make NMRU look like a “clear winner” in Table 3? Throughout the discussion of the results and looking at the plots I had a sense that there are some small, but clear improvements from NMRU, but an overall still quite mixed story; whereas Table 3 gives the impression that NMRU is a massive improvement over the other two. For instance: in Fig 7, RealNPU passes the 75% threshold for one large-magnitude uniform task but not the other, such that it does not pass overall (though clearly it does perform worse than NRU/NMRU). It would be helpful if there was more systematicity to how the ranges are selected. Or perhaps it would also help a lot to report numbers in Table 3 and simply make the entries that currently have a check-mark bold. Since this is not a method paper, but mainly an empirical analysis, I think it is more important to give a differentiated overview over strengths and weaknesses, rather than producing a clear winner method.
3. It would be nice to see NRU/NMRU also evaluated on the tasks in Fig. 4. Any reason to exclude them?
4. The paper could be better organized. It would help to describe the tasks (somewhat) before the models, and even within the models I feel like 3.1 is much harder to follow than 3.2 and 3.3 (perhaps also move 3.1 after 3.3?).
5. The bibliography seems very short (unless I missed something I am counting 9 directly related references and 13 references in total; the last one is a duplicate) and the related work discussion could go into a bit more details about the differences and similarities. While there is no need for dozens of references, and I am not expert enough to point out which ones are missing, it still seems to me that a more comprehensive literature review might be beneficial.
6. More detailed and differentiated discussion. My takeaway from the results is that RealNPU performs fairly broadly but most often a bit worse than NRU (when it works) and NMRU. NRU has a systematic shortcoming as shown in Fig 6, and NMRU is also not completely free of issues. In particular, it is still a bit unclear that NMRU is a practical alternative to RealNPU, since NMRU requires double the input dimensionality and, crucially, inversion of the whole input (even the irrelevant parts). This makes NMRU scale less well. It would be nice to discuss all of this with a bit more detail.
7. Throughout the paper I would have been keen to see the percentage of non-sparse or non-discrete parameters (what percentage lies more than some epsilon away from the nearest discrete value, i.e. -1, 0, or 1?). This would be an interesting additional measure, perhaps even more insightful than the sparsity error.

Minor (not crucial for acceptance)

There is a fairly large body of literature on learning neural networks with binary or ternary weights, and pruning weights (sparsity) in the field of neural network compression. It might be worth (for the future) to have a look at the state-of-the-art for further improvements a la section 5.

Beginning of Sec. 4: explain sparsity regularisation scaling and why it is needed (it can be inferred from context, and will be clear to an expert reader, but adding a sentence would not hurt)?

How were the different iterations for switching on/off the sparsity regularisation determined?

Table 1 is missing some labeling or description: what are the different rows, what is the difference between rows 1, 2 and 3, 4?

Results in Sec. 5: There is a succession of ablations; does each subsequent setting use the best outcome from the previous ablation? E.g. all later experiments use L1 regularization with the best range, all experiments after Fig. 2a use GW clipping, etc?

Fig. 3, panel 3: The sparsity error is dominated by two outliers, which makes it really hard to see what happens in the rest of the plot (the scaling of the y-axis is too large). Consider providing a plot without RealNPU and/or a log-scale y-axis in the main paper.

A few typos/misspellings throughout (e.g. diving instead of dividing on P7, effects instead of affects on P9, …). Make sure to do another pass for errors/typos.

Wall-clock timings in Table 7 cannot be meaningfully compared when experiments are run on very different machines.

Have you experimented with a “more held-out” validation set (that already requires some extrapolation), rather than an i.i.d. validation set?


**Strengths And Weaknesses:**

Pro
 * The paper introduces four sensible improvements to the baseline method and evaluates these through thorough ablation studies.
 * The paper introduces two novel neural modules for division, which are simple but theoretically sound, and evaluates their performance through a battery of simple experiments.
 * Results are consistently computed from 25 repetitions over random seeds, with appropriate statistics, which leads to highly reliable comparisons.

Con:
 * To make the paper accessible to a wider audience, it lacks a more general introduction and explanation of the goals and challenges with Neural ALUs. This is fine for an expert audience, but I think there is some room for improvement.
 * It is unclear how experimental settings were designed. One experiment consists of a certain distribution over inputs (for training and validation set), and another non-overlapping distribution over inputs for the test set (typically implying larger-magnitude values than the training set). There are two concerns with this:
   * Can it be ruled out that experiments were not picked to favor the novel method over the modified baseline, in particular NMRU (indirect cherry picking to tick as many lines as possible in Table 3)? This is a minor concern since some experiments show shortcomings of the proposed method (which is good for any follow-up work). See Requested Changes for more discussion.
   * The main story of the paper could perhaps be told in a more compact fashion by selecting a subset of all experiments in the paper and using them as the primary example. For instance, why not have one set of experiments that include mainly mixed-sign ranges with two-sided generalization (extrapolation on both sides of the interval)? It seems to me like the latter is the main experiment, the single-sign ranges with one-sided generalization can sometimes offer additional info, but overall Table 1 seems to have too many redundant settings, and not enough interesting ones (which are presented later). See Requested Changes for more discussion.
 * At the end of the paper it is still unclear whether the way forward is to come up with more, or better, modifications of the baseline (or NRU/NMRU), or whether a new Neural Division Model is needed to address the shortcomings shown in the paper. This is, of course, not an easy task - but it makes the current findings somewhat inconclusive (which is still nice for future work building on these results, but the paper would certainly be stronger with a stronger answer to this question).

---

> ### Author Response · Authors · 2022-06-21
> **Response to reviewer fzJX - #1 (part 1)**
>
> Firstly, thank you for your fast and detailed reply. We have taken on the suggested improvements and will provide the discussed changes (see below) in the revised paper, which will be uploaded once all reviewers have responded. The following will respond to the requested changes with any references found at the very end. (Note that due to character limitations the full response is over multiple comments.)
>
> In response to your requested changes:
>
> **Major/important changes**
>
> Perhaps the main experiments can be distilled down into a smaller, more informative, set with the remaining ones moved to the appendix.
> Why does Table 1 only have a single mixed-sign experiment and only a single two-sided extrapolation experiment?
>
> - The chosen ranges are directly influenced by prior works, see [1] and [2], in which the authors only used a single mixed-sign experiment and a single two-sided extrapolation experiment. We do explore additional mixed sign datasets (see Figure 4) but as you pointed out this is currently only done on the Real NPU.
>
> I would have liked to see more of the datasets that went into Figure 4 and Table 2 as the main experiment (and only one or two representative single-sign, one-sided extrapolation experiments).
> - For the revised paper we will add the results of the NRU and NMRU on the mixed signed dataset (figure 4).
> - We will update the paper to empathise more of the results from Figure 4 and Table 2.
> - We believe that the one-sided extrapolation ranges are relevant to the main story as they depict the effect of training on ranges close to and not close to the singularity point (of 0).
>
> Cherry-picking: were some additional experiments selected to make NMRU look like a “clear winner” in Table 3?
> - We only use the experiments referenced in the “Figure” column to determine the tables results. However, we agree that currently Table 3 can be misleading in that it seems to favour the NMRU, so we will modify the Table (see the question about reporting numbers below).
>
> It would be helpful if there was more systematicity to how the ranges are selected.
> - We will add text to clarifying the ranges used to in Table 3. In particular, the ranges for Table 3 are selected in order to highlight the areas which the modules stuggle (i.e., mixed-signs, negative ranges, large ranges and ranges close 0) to indicate the types of data/distributions to focus on improving.
>
> Or perhaps it would also help a lot to report numbers in Table 3 and simply make the entries that currently have a check-mark bold.
> - We agree this would provide more value to the reader. We will replace the checks with numbers and provide an indication of the best model for each row (e.g., via colour/bold).
>
> It would be nice to see NRU/NMRU also evaluated on the tasks in Fig. 4. Any reason to exclude them?
> - We had not included the NRU/NMRU initially because the focus was towards the Real NPU, as it was the only model to struggle on a mixed sign range (see Figure 3). In the revised paper we will add their results aswell (with results showing that neither module struggled with the mixed-sign ranges).
>
> The paper could be better organized. It would help to describe the tasks (somewhat) before the models, and even within the models I feel like 3.1 is much harder to follow than 3.2 and 3.3 (perhaps also move 3.1 after 3.3?).
> - At the start of section 3 we will add a description of the tasks and relate them to the NALMs. We choose to have 3.1 first as it refers to an existing module unlike the NMU and NMRU which are novel modules, hence it seemed more natural to start with. Perhaps adding illustrations of the module architectures could help understand the mathematical definitions better?
>
> The bibliography seems very short (unless I missed something I am counting 9 directly related references and 13 references in total; the last one is a duplicate) and the related work discussion could go into a bit more details about the differences and similarities.
> - Can you clarify if the further discussion should be towards the NALM literature, the general symbolic regression literature, or both?
> - The field for arithmetic units is fairly recent, which is why there are not many direct works to reference.
> - Thank you for pointing out the duplicate citation; we will fix this in the revised paper.
>
> ...

---

> > ### Author Response · Authors · 2022-06-21
> > **Response to reviewer fzJX - #1 (part 2)**
> >
> > ...
> >
> > More detailed and differentiated discussion.
> > My takeaway from the results is that RealNPU performs fairly broadly but most often a bit worse than NRU (when it works) and NMRU.
> > NRU has a systematic shortcoming as shown in Fig 6, and NMRU is also not completely free of issues.
> > In particular, it is still a bit unclear that NMRU is a practical alternative to RealNPU, since NMRU requires double the input dimensionality and, crucially, inversion of the whole input (even the irrelevant parts). This makes NMRU scale less well. It would be nice to discuss all of this with a bit more detail.
> > - The discussion section will be extended to differentiate these points better to provide a stronger conclusion. In particular, more discussion in highlighting the NMRU scaling and parameter count issues will be given.
> >
> > Throughout the paper I would have been keen to see the percentage of non-sparse or non-discrete parameters (what percentage lies more than some epsilon away from the nearest discrete value, i.e. -1, 0, or 1?). This would be an interesting additional measure, perhaps even more insightful than the sparsity error.
> > - In the revised paper, we will include this metric for the input-size 10 task on the more challenging distributions.
> >
> > **Minor (not crucial for acceptance)**
> >
> > There is a fairly large body of literature on learning neural networks with binary or ternary weights, and pruning weights (sparsity) in the field of neural network compression.
> > - Thank you for the suggestion. We will mention this as a direction for future work.
> >
> > Beginning of Sec. 4: explain sparsity regularisation scaling and why it is needed (it can be inferred from context, and will be clear to an expert reader, but adding a sentence would not hurt)?
> > - An explanation will be added in the revised copy.
> >
> > How were the different iterations for switching on/off the sparsity regularisation determined?
> > - The iterations are chosen to allow weights to have the opportunity to explore and begin to move far enough to the ideal discrete value (within 0.5) which is why the earliest regularisation is only switched on after 50% of the epochs have occurred. From our results, we have shown this works empirically.  Furthermore, as the warmup scaling used for the regularisation occurs over many epochs meaning there is some leniency in the epochs to start and end the scaling without significantly impact success. In some cases, it could be possible to tune the start and end epoch for faster convergence but this was out of the scope for this study and left for future work.
> >
> > Table 1 is missing some labeling or description: what are the different rows, what is the difference between rows 1, 2 and 3, 4?
> > - Rows 3 and 4 is a result of wrapping Table 1 (since putting all 9 ranges in 1 row is too long). Rows 1 and 3 are the interpolations ranges and rows 2 and 4 are the respective extrapolation ranges to the interpolation range shown in rows 1 and 3. E.g., For interpolation range [1,2) you have extrapolation range [2,6).
> >
> > Results in Sec. 5: There is a succession of ablations; does each subsequent setting use the best outcome from the previous ablation?
> > - Yes it does. We mention in Section 5: ‘To address each question in order, we propose applying incremental modifications to the Real NPU.’ But if this is still unclear, we can reword it.
> >
> > Fig. 3, panel 3: The sparsity error is dominated by two outliers, which makes it really hard to see what happens in the rest of the plot (the scaling of the y-axis is too large). Consider providing a plot without RealNPU and/or a log-scale y-axis in the main paper.
> > - We will look into providing this plot this for the revised paper.
> >
> > A few typos/misspellings throughout (e.g. diving instead of dividing on P7, effects instead of affects on P9, …). Make sure to do another pass for errors/typos.
> > - Thank you for letting us know. We will do another proof read.
> >
> > Wall-clock timings in Table 7 cannot be meaningfully compared when experiments are run on very different machines.
> > - Table 7 is given as using approximate times to give the reader an idea of experiment run times if they are interested in reimplementation. These times should be used as a simple indicator rather than exact timings.
> >
> > Have you experimented with a “more held-out” validation set (that already requires some extrapolation), rather than an i.i.d. validation set?
> > - We do not because the validation set is used for early stopping meaning the data should not leak any possible information about the extrapolation range. The validation set should be seen as testing the model on the interpolation data, however as the focus of this paper is extrapolation we report the results on the extrapolation sets.
> >
> > **Extra points**
> >
> > - In response to your concern about the Introduction (mentioned in the cons section of your reply), we will look at rewording the Introduction to introduce Neural ALUs (NALMs) earlier.
> >
> > ...

---

> > > ### Author Response · Authors · 2022-06-21
> > > **Response to reviewer fzJX - #1 (part 3)**
> > >
> > > **References:**
> > >
> > > [1] Andreas Madsen, and Alexander Rosenberg Johansen 2019. Measuring Arithmetic Extrapolation Performance. In Science meets Engineering of Deep Learning at 33rd Conference on Neural Information Processing Systems (NeurIPS 2019).
> > >
> > > [2] Andreas Madsen, and Alexander Rosenberg Johansen 2020. Neural Arithmetic Units. In International Conference on Learning Representations.

---

> > ### Comment · Reviewer_fzJX · 2022-07-17
> > **Thank you for the detailed response**
> >
> > Thank you for clarifying and incorporating many changes, including additional evaluation. I am happy with all the changes and do not think that any major points preventing publication are left open (I sympathise with some comments by 3o96, and agree that showing that the modules can learn division "in the wild" would be impressive; however as the current results suggest, there is more work to be done (in the community) before we can hope to tackle that problem - so I think there is some merit in publishing the current results, though they are of more limited significance to the wider community; significance and impact are not major concerns for this outlet though).
> >
> > Clarifications/comments:
> >
> > Re: organization of section 3. I think that logically it makes sense to indroduce the Real NPU first - but in the previous manuscript I found that subsection hardest to follow, and found some answers to high-level questions I had w.r.t. 3.1 in Section 3.2 and 3.3. I think this has been sufficiently addressed, and a bit more intro to neural ALUs (NALM) is certainly helpful to make the paper more easily accessible to non-experts.
> >
> > Re: bibliography/literature. I understand that the field is relatively new, and as a non-expert would have expected a bit more relevant literature over the last years. But as I said, I could not point out anything that's missing. I do think adding a short more general NALM intro is beneficial (compared to the originally submitted manuscript). So no need to add unnecessary references - this was perhaps just a reminder to think whether all the important ones are there.
> >
> > Re results in Sec 5 being consecutive improvements. Thank you I missed that sentence (despite briefly looking for it); no need to change the wording.

---

### Review · Reviewer_3o96 · 2022-06-29

**Summary Of Contributions:**

This paper explores simple neural modules for learning division operations. The goal is to develop interpretable units that could be included in neural networks trained with gradient descent that could learn to perform division in a manner that generalizes outside of the training range, even when there are multiple inputs, including irrelevant ones. The authors design two novel units (the NRU and NMRU), and they explore the performance of these units (as well as an older one, the Real NPU) across a variety of domains and task set-ups for explicit division, and with various ablations and manipulations of the parameters and components of the units. Their results provide clear evidence that the NRU and NMRU work better than the Real NPU on the sort of tasks explored.

**Broader Impact Concerns:**

I do not have any concerns about the broader impact of this work.

**Requested Changes:**

1) This work is lacking a major piece of data, namely, any kind of demonstration that these units learn division when they are part of a larger network learning some other task. Without any data of this nature the work is not convincing as to the technical capabilities of these units in the sort of applications one would actually want to use them on. Really, to make this work relevant to other researchers, the authors would show some non-trivial learning of division as a result of network modelling of a known ground truth system that includes division as a key operation.

2) There are a number of areas where the clarity of the paper could be improved:

- In equation 3, is g a function of the input (as would be the case in, e.g. an LSTM), or literally just an extra set of fixed parameters?
- What exactly is meant by a "weighted L1 loss"? There are a couple of possible interpretations of this, but it appears that here there is a single weight, so do the authors simply mean that the add an L1 loss whose impact is weighted by beta? Also, why is a schedule for this weight necessary?
- How exactly are the various weights and parameters clipped to remain in range? Do you use simply set them to the max/min whenever those values are passed?
- In equation 4, is o simply an index for the output unit? Why include that in the absence of any tests with multiple outputs?
- How did you choose the scale factor of 1000 for the tanh version of absolute value?
- Why bother using cos to do sign calculations? Is that more efficient than simply multiplying the signs of the input elements together?
- What do the different columns in Table 1 correspond to?
- How were the default parameters selected?
- Why use the ranges of Madsen & Johanson (2020)
- Typo page 4: "Sparsity error (is) calculated by:"
- Why use parametric distributions to calculate confidence intervals? With only 25 seeds one could easily just show the data. Alternatively, one could use a non-parametric bootstrap method to calculate the CIs.
- On page 6 the scaling factor is mentioned before being introduced.
- Bottom of page 6: what modifications to the NPU are used here, exactly?
- Typo page 8: "10 input task(s)"
- Typo page 9: "regardless (of) the value"
- Typo page 11: "the use*d* of mixed signed inputs"
- Figure 9: the text is too small too read. I know the figures are presented in larger form in the Appendix, but still, it's not good practice to include any figures where one cannot read the text.
- Table 5: similar to above, how were these parameters chosen?
- Equation 12: what is the "clamp" operation here, exactly?


**Strengths And Weaknesses:**

Strengths:

1) The authors do a thorough investigation across ranges and ablations.
2) The new units proposed do clearly work better than the older Real NPU variant on the tasks tested here.

Weaknesses:

1) There is a big disconnect between the potentially desired uses of these division units and the tests performed in this paper. While reading this paper, I was left asking myself, why would anyone want to train these modules in these ways? To unpack that statement: the motivation for having a neural unit that can learn division is surely so that it can be included in a larger neural network being trained on more complex, interesting tasks. If we were modelling some system (be it some physics problem, or whatever), and our network included these units, and training to match the system being studied led these units to learn division of certain variables, then we would potentially have some interpretable description of the system. Great! But, I can think of no situation in which one would actually want to train these units in isolation on literal division tasks as the authors do here. Now, the intuition may be that if we cannot show that the units learn division when explicitly trained to do so in isolation then we certainly can't be confident that they would work when trained as part of a larger network. That may be, but it's hard to say - networks behave very differently than individual units and as the authors' own data shows that if a different loss is used it can also effect training. Moreover, we also cannot say that if these units work when trained explicitly to do division in isolation that they will then learn division when trained as part of a larger network. As such, the work here essentially demonstrates that these units work better on a series of tasks that no one would ever actually want to use them for, and doesn't provide any evidence that they work well on the sorts of tasks that one might actually want to use them for. This means that the technical prowess of this system for the actual desired uses is not demonstrated at all.

2) The clarity of the exposition is weak in a few places. It took me a while to fully understand the various components, a number of parameter choices were opaque, and some of the tables and figures are hard to interpret. I will provide details of this below.

---

> ### Author Response · Authors · 2022-07-06
> **Response to Reviewer 3o96 - #1 (part 1)**
>
> Thank you for your in-depth response and suggestions. We have taken on the suggested improvements and will provide the discussed changes (see below) in the revised paper, which will be uploaded soon. Below, we respond to your requested changes. Please assume anything points regarding typos/ formatting issues will be fixed in the paper revision.  (Note that due to character limitations the full response may be over multiple comments.)
>
> **Weaknesses:**
>
> Now, the intuition may be that if we cannot show that the units learn division when explicitly trained to do so in isolation then we certainly can't be confident that they would work when trained as part of a larger network. That may be, but it's hard to say - networks behave very differently than individual units and as the authors' own data shows that if a different loss is used it can also effect training. Moreover, we also cannot say that if these units work when trained explicitly to do division in isolation that they will then learn division when trained as part of a larger network.
> - Thank you for raising this point. Our work does use the assumption that if a module has the capability to work on something complex (such as being attached to a deeper model) then it should also work on the simpler cases as well and therefore are worth exploring. The advantage of the simpler setup is that we can control the independent variables without having to worry about the effect of the learning of other components in a network. As you mentioned, learning in isolation and with other networks will have differences but we believe it is also important to understand the learning of the modules in the simple cases as well. For example, we have discovered that input redundancy makes the NRU difficult to learn, mixed-sign inputs cause issues with the Real NPU and large magnitudes can cause the NMRU to struggle.
> - Our intention of showing the effect of different losses is to raise awareness for researchers on an additional factor (which can be forgotten about) that has direct impact on learnability. Since work on these division modules is only in its infancy, we want to provide intuitions for researchers to build on.
>
> As such, the work here essentially demonstrates that these units work better on a series of tasks that no one would ever actually want to use them for, and doesn't provide any evidence that they work well on the sorts of tasks that one might actually want to use them for. This means that the technical prowess of this system for the actual desired uses is not demonstrated at all.
> - Again, we empathise that the purpose of the synthetic experiments is to identify areas of failure and determine ways in which we can alleviate these failures and validate them empirically. The contributions of this paper should not be viewed as irrelevant but a result of a much more rigorous analysis, provided by our range of experimental results and theoretical findings, acting as a necessary step in understanding the learning dynamics of these modules such that researchers can better utilise them in future work.
>
> **Requested Changes:**
>
> This work is lacking a major piece of data, namely, any kind of demonstration that these units learn division when they are part of a larger network learning some other task. Without any data of this nature the work is not convincing as to the technical capabilities of these units in the sort of applications one would actually want to use them on. Really, to make this work relevant to other researchers, the authors would show some non-trivial learning of division as a result of network modelling of a known ground truth system that includes division as a key operation.
> - Thank you for the suggestion. To test the modules on a more challenging task, we are currently running experiments for a regression task requiring an intermediary classification network consisting of a ConvNet.
>
> In equation 3, is g a function of the input (as would be the case in, e.g. an LSTM), or literally just an extra set of fixed parameters?
> - The g is a learnt vector of parameters, currently explained in the paper as: “A gating vector g, learns to select relevant input elements, where gate values are clipped between [0,1] during training.”
>
> What exactly is meant by a "weighted L1 loss"? There are a couple of possible interpretations of this, but it appears that here there is a single weight, so do the authors simply mean that the add an L1 loss whose impact is weighted by beta? Also, why is a schedule for this weight necessary?
> - The weighted L1 loss is used by the authors of the Real NPU and is simply the L1 regularisation penalty which is scaled (i.e. weighted) by some factor (beta). The schedule can be interpreted as a way of gradually increasing the impact of the L1 penalty. We use this scheme for the (Real) NPU to replicate how the module was trained by the original authors.

---

> > ### Author Response · Authors · 2022-07-06
> > **Response to Reviewer 3o96 - #1 (part 2)**
> >
> > ...
> >
> > How exactly are the various weights and parameters clipped to remain in range? Do you use simply set them to the max/min whenever those values are passed?
> > - During the forward pass of the modules the relevant parameters are clipped such that each element of the parameter is within the allowed min/max range. Therefore when the modules apply their logic the learnable parameters would have been clipped to the allowed range. E.g., for the gate vector for the Real NPU with values [-0.1, 0.4, 0.7, 1.1] would be clipped between [0,1] resulting in the clipped vector: [0, 0.4, 0.7, 1].
> > - Specifically, we use the pytorch method torch.clamp() (see https://pytorch.org/docs/stable/generated/torch.clamp.html?highlight=clamp#torch.clamp) to implement the clamping.
> >
> > In equation 4, is o simply an index for the output unit? Why include that in the absence of any tests with multiple outputs?
> > - Yes, “o” is the index for the output. We use this notation because we want to display the general form of the functions. We realise the NPU equations are not given in this form, so for consistency we shall update it to do so.
> >
> > How did you choose the scale factor of 1000 for the tanh version of absolute value?
> > - In preliminary experiments, we found that tanh with a large scaling factor 1000 performs the best, which correlates with the findings in Faber & Wattenhofer’s (2020)) Figure 5. We can add the results to the Appendix if desired.
> >
> > Why bother using cos to do sign calculations? Is that more efficient than simply multiplying the signs of the input elements together?
> > - Using no sign mechanism by letting the inputs retain their signs and multiply directly results in worse performance as shown by out ablation study in Figure 17 (compare gnc (NMRU with no sign) to gnc+sign (NMRU).
> >
> > What do the different columns in Table 1 correspond to?
> > - Table 1 shows the different interpolation ranges used for the train and validation set and the corresponding extrapolation range used for testing e.g., [-20, -10) and [-40,-20). In total there are 9 different ranges shown.
> >
> > How were the default parameters selected?
> > - We use the parameters found from existing works. For example the RealNPU scaling parameters come from Heim et al. (2020) and the discretisation parameters and experiment setup parameters (e.g. optimizer) follow the work on the NMU by Madsen & Johanson (2020).
> >
> > Why use the ranges of Madsen & Johanson (2020)
> > - We believe that the Madsen & Johanson (2020) ranges provide a sensible set of ranges with good coverage which are designed not to be biased for testing division. Furthermore, as their work is published in a reputable conference (i.e., ICLR) it implies it has been considered a reasonable choice of ranges by others as well.
> >
> > Why use parametric distributions to calculate confidence intervals? With only 25 seeds one could easily just show the data. Alternatively, one could use a non-parametric bootstrap method to calculate the CIs.
> > - We choose these parametric distributions for consistency. We follow the evaluation metrics specified by Madsen & Johanson (2019) as we believe it’s important to be consistent in the evaluation metrics to allow comparability between methods.
> >
> > Bottom of page 6: what modifications to the NPU are used here, exactly?
> > - They refer to the modifications made in Section 5. I.e., clipping, discretisation regularisation and a constrained uniform Xavier initialisation of the real weight matrix.
> >
> > Figure 9: the text is too small too read. I know the figures are presented in larger form in the Appendix, but still, it's not good practice to include any figures where one cannot read the text.
> > - We will update the paper to either have larger images in the main body of the text or remove the images completely from the main body.
> >
> > Table 5: similar to above, how were these parameters chosen?
> > - We use the parameters used in Heim et al. (2020, Section 4.1) which we confirm empirically in Figure 1b.
> >
> > Equation 12: what is the "clamp" operation here, exactly?
> > - The clamp sets the lower bound to epsilon. We will make this more clear.

---

> > > ### Comment · Reviewer_3o96 · 2022-07-26
> > > **Very close to ready**
> > >
> > > Thank you to the authors for their excellent responses to my concerns. My second major concern is fully addressed.
> > >
> > > As to my first major concern, the new MNIST data is precisely the type of data I was hoping to see, so well done. I think, though, given those results, the authors need to add some more discussion to the paper. Specifically, the MNIST data shows the Real NPU outperforming the NRU and NMRU, despite having under-performed in the other tests on basic division! This speaks to exactly my original concern, namely, that performing well on division as a single unit does not necessarily imply good performance when embedded in a larger network.
> > >
> > > Currently, the authors merely note that "...robustness of the NALMs (such as the NMRU) require further improvements". I think they need to go a couple of steps farther here. Specifically, they need to be open with readers that this data shows (1) that the results on single layer division tasks cannot be taken as a guarantee that NRUs and NMRUs will outperform Real NPUs when embedded in larger networks, and (2) that anyone who wants to use these units in some other network will need to perform their own tests in the target domain before employing one of these units.
> > >
> > > If the authors can expand this important cautionary message for readers, than I am happy to recommend acceptance.

---

> > > > ### Comment · Reviewer_UuYH · 2022-07-27
> > > > **Agree with 3o96**
> > > >
> > > > I agree with R-3o96. While the authors have already taken care not to make overly broad claims, expanding on the conclusions with important caveats stated clearly will improve the paper.

---

> > > > ### Author Response · Authors · 2022-07-27
> > > > **Response to 3o96 and UuYH - additional discussion on the MNIST task**
> > > >
> > > > In our next paper update, we will discuss the two points mentioned above by Reviewer 3o96. The next paper update will be uploaded by the 1st of August (01/08/22) latest. Thank you again for your responses.

---

### Review · Reviewer_UuYH · 2022-07-06

**Summary Of Contributions:**

This paper is concerned with the problem of training neural networks to learn the division operation using gradient descent. The motivation is that using the ability to represent precise mathematical functions will be useful for training neural networks in situations where there is aprecise mathematical relationships between variables in the data, such as certain physical systems. Another potential use of such modules is in learning interpretable functions of the inputs (such as controllers).

The paper builds on two prior ideas: Real Neural Power Unit (Real NPU), an existing module that can learn multiplication and addition,  and Neural Multiplication Unit (NMU), a module for learning multiplication. For the Real NPU, it discusses and experimentally demonstrates its failure modes when learning division, and demonstrates that techniques based on the NMU paper can be used to substantially improve its performance and robustness. The paper takes inspiration from NMU to propose two new modules for learning division: Neural Reciprocal Unit (NRU) and Neural Multiplicative Reciprocal Unit (NMRU). The key insight is that if we have a module that is effective at learning multiplication, then it can be extended to learn division easily if a) the ability to take reciprocals is added to the module (NRU), or b) the reciprocal of each input is additionally provided as an input to the module (NMRU).

The main contribution is a detailed experimental study evaluating the Real NPU, Real NPU with proposed modifications, NRU, and NMRU under a large variety of training/test set ranges. The experiments are designed to evaluate robustness to different training ranges, ability to extrapolate to different testing ranges, and ability to learn to divide relevant inputs while ignoring irrelevant inputs. In addition, there are further experiments focused on specific challenges and failure cases for learning division.

Overall, the Modified Real NPU and NMRU are identified as promising modules for future work. Failure cases for both of these modules are discussed, which are mainly related to mixed-signed inputs, large input ranges and redundant inputs.

**Requested Changes:**

None of the following changes are critical.

- Indicate how the learning rate of Real NPU was tuned and whether the measured performance is indicative of the best that can be achieved with reasonable tuning.
- Fix typo in pg. 1, second to last para: "differentable"
- Fix extra space after "Section 4.1)" in Sec. 5
- Add a table of contents to the appendices since there are several sections


**Strengths And Weaknesses:**

This paper's main contribution is also its main strength: a detailed experimental study that compares the studied modules (Modified Real NPU, NRU and NMRU) on a diverse range of synthetic test problems. The problems differ on two main axes: one is the range of inputs used for training/validation and test sets, and the other is the "no redundancy" setting with 2 inputs (where the task is learn to divide one input by the other) vs. "with redundancy" setting with 10 inputs (where the network should also learn to ignore the 8 irrelevant inputs). With this coverage, despite the synthetic nature of the tasks, I find that the evaluation is sufficient to understand the strengths and failure modes of the various models.

The NRU/NMRU are very intuitive extensions of the NMU, and "why can't the NMU be extended to division?" seems to be a very natural question to ask. This experimental study fills a clear gap in the literature by exploring this possibility.

Another strength of the paper is that it thoroughly evaluates the importance of choices related to training, regularization and initialization in addition to the model design. Based on the results, it can be observed that these choices may in many cases be more important than the model design itself. It also closely examines a series of specific issues related to learning division in detail such as gradient difficulties with the NRU, exploitation of multiplicative rules, and extrapolation to large ranges. Through these experiments, I believe that this paper sets a good standard for future work.

I did not find any major weaknesses in this study. A small concern is whether the hyperparameters for Modified Real NPU (mainly the learning rate of 5e-3) were tuned, similar to NRU/NMRU (based on results in appendix), or not. Sec. 5 does indicate that the L1 regularization hyperparameters were tuned.

---

> ### Author Response · Authors · 2022-07-06
> **Response to reviewer UuYH - #1**
>
> Thank you for your comments. The learning rate for the Real NPU was initially taken from the original paper by Heim et al. (2020) but we also have results to show that this was an empirically valid choice in comparison to other learning rates. We will add the results to the appendix. The additional points for formatting and typo fixes will also be added.

---

### Author Response · Authors · 2022-07-28
**Paper revision 2 - additional discussion on the MNIST task**

 To reviewers 3o96 and UuYH. The paper has been revised to include the 2 points of caution (see changes since last submission).

---

> ### Comment · Reviewer_3o96 · 2022-07-28
> **Good to go**
>
> The revised paper addresses my final concerns, thank you. I will now recommend acceptance.

---

> ### Comment · Reviewer_UuYH · 2022-07-28
> **Thank you**
>
> Thanks for the update, and for the insightful paper.

---

### Decision · Action_Editors · 2022-08-11

**Recommendation:** Accept as is

**Comment:**

**Paper Summary**

This paper focuses on different types of neural units for learning division with backpropagation and gradient descent.  The paper empirically evaluates the Real NPU, Real NPU with proposed modifications, NRU, and NMRU across a large variety of datasets, tasks and domains. The main contribution of the paper is the thorough experiments, and ablations of different neural modules for learning division operation.

**Decision**

Overall, the authors did a very good job in the rebuttal phase and successfully addressed the concerns raised by the reviewers. The paper is well-written, clear and well-motivated. The experiments are very thorough and evaluation protocol is carefully thought. The authors seem to provide sufficient details about their experiments to be able to reproduce their results. The reviewers agreed that the findings of this paper can be beneficial for the TMLR and broader machine learning community and the authors incorporated the changes requested by the reviewers into the paper. In a nutshell, this paper is ready to be published at TMLR in this form.